# Inflammatory response in hematopoietic stem and progenitor cells triggered by activating SHP2 mutations evokes blood defects

**Maja Solman[1], Sasja Blokzijl-Franke[1†], Florian Piques[2,3†], Chuan Yan[4,5,6,7], Qiqi Yang[4,5,6,7], Marion Strullu[2,8], Sarah M Kamel[1], Pakize Ak[1], Jeroen Bakkers[1,9], David M Langenau[4,5,6,7], Hélène Cavé[2,3], Jeroen den Hertog[1,10]***

[1]Hubrecht Institute-KNAW and UMC Utrecht, Utrecht, Netherlands; [2]INSERM UMR_S1131, Institut de Recherche Saint-Louis, Université de Paris, Paris, France; [3]Assistance Publique des Hôpitaux de Paris AP-HP, Hôpital Robert Debré, Département de Génétique, Paris, France; [4]Molecular Pathology Unit, Massachusetts General Hospital Research Institute, Charlestown, United States; [5]Massachusetts General Hospital Cancer Center, Charlestown, United States; [6]Center for Regenerative Medicine, Massachusetts General Hospital, Boston, United States; [7]Harvard Stem Cell Institute, Cambridge, United States; [8]Assistance Publique des Hôpitaux de Paris AP-HP, Hôpital Robert Debré, Service d'Onco-Hématologie Pédiatrique, Paris, France; [9]Department of Medical Physiology, Division of Heart and Lungs, UMC Utrecht, Utrecht, Netherlands; [10]Institute of Biology Leiden, Leiden University, Leiden, Netherlands

*For correspondence:
j.denhertog@hubrecht.eu

†These authors contributed equally to this work

Competing interest: The authors declare that no competing interests exist.

**Abstract** Gain-of-function mutations in the protein-tyrosine phosphatase SHP2 are the most frequently occurring mutations in sporadic juvenile myelomonocytic leukemia (JMML) and JMML-like myeloproliferative neoplasm (MPN) associated with Noonan syndrome (NS). Hematopoietic stem and progenitor cells (HSPCs) are the disease propagating cells of JMML. Here, we explored transcriptomes of HSPCs with SHP2 mutations derived from JMML patients and a novel NS zebrafish model. In addition to major NS traits, CRISPR/Cas9 knock-in Shp2$^{D61G}$ mutant zebrafish recapitulated a JMML-like MPN phenotype, including myeloid lineage hyperproliferation, ex vivo growth of myeloid colonies, and in vivo transplantability of HSPCs. Single-cell mRNA sequencing of HSPCs from Shp2$^{D61G}$ zebrafish embryos and bulk sequencing of HSPCs from JMML patients revealed an overlapping inflammatory gene expression pattern. Strikingly, an anti-inflammatory agent rescued JMML-like MPN in Shp2$^{D61G}$ zebrafish embryos. Our results indicate that a common inflammatory response was triggered in the HSPCs from sporadic JMML patients and syndromic NS zebrafish, which potentiated MPN and may represent a future target for JMML therapies.

## Editor's evaluation

The authors of this paper model the D61G mutation in the gene PTPN11 that encodes the protein-tyrosine phosphatase SHP2 in zebrafish, creating a model consistent with the human Noonan syndrome (NS), which is predisposed to juvenile myelomonocytic leukemia (JMML) and myeloproliferative neoplasm (MPN)-like syndrome. The study nicely provides a new model that can be used as the basis for future studies in the field. Because the mutant variably displays phenotypes along a spectrum from NS to MPN, different researchers can choose to focus on this as they see fit.

**eLife digest** Juvenile myelomonocytic leukaemia is a childhood blood cancer. It is more common in children with a genetic condition called Noonan Syndrome, which causes problems with development in many parts of the body. The most frequent cause is a mutation in a protein called Src homology region 2 domain-containing phosphatase-2, or SHP2 for short.

Juvenile myelomonocytic leukaemia starts in the stem cells that normally become blood cells. In children with Noonan Syndrome, these cells show signs of problems before leukaemia begins. Recreating Noonan Syndrome in an animal could shed light on how this childhood cancer develops, but doing this is not straightforward. One option is to use zebrafish, a species of fish in which the embryos are transparent, allowing scientists to watch their blood cells developing under a microscope. They also share many genes with humans, including SHP2.

Solman et al. genetically modified zebrafish so they would carry one of the most common mutations seen in children with Noonan Syndrome in the SHP2 protein. The fish had many of the typical features of the condition, including problems producing blood cells. Single cell analysis of the stem cells that become these blood cells showed that, in the mutated fish, these cells had abnormally high levels of activity in genes involved in inflammation. Treating the fish with an anti-inflammatory drug, dexamethasone, reversed the problem. When Solman et al. investigated stem cells from human patients with juvenile myelomonocytic leukaemia, they found the same high levels of activity in inflammatory genes.

The current treatment for juvenile myelomonocytic leukaemia is a stem cell transplant, which is only successful in around half of cases. Finding a way to prevent the cancer from developing altogether could save lives. This new line of zebrafish allows researchers to study Noonan Syndrome in more detail, and to test new treatments. A next step could be to find out whether anti-inflammatory drugs have the same effects in mammals as they do in fish.

## Introduction

A broad spectrum of heterozygous germline activating mutations in the tyrosine phosphatase SHP2 encoded by *PTPN11* has been found to cause Noonan syndrome (NS), a dominantly inherited developmental disorder from the RASopathy group affecting 1:1500 individuals. NS is characterized by a systemic impact on development, most commonly resulting in short stature, congenital heart defects, and specific craniofacial characteristics (*Rauen, 2013*; *Tajan et al., 2018*). Somatic activating mutations in *PTPN11* are the most common cause of sporadic juvenile myelomonocytic leukemia (JMML), a rare but aggressive myelodysplastic and myeloproliferative neoplasm (MPN) occurring in young children (*Caye et al., 2015*; *Tartaglia et al., 2003*). Consistently, children with NS are predisposed to developing neonatal MPN, that either regresses without treatment or rapidly progresses to JMML leading to early death (*Kratz et al., 2005*; *Strullu et al., 2014*). Previous studies support the view of a strong endogenous role of germline *PTPN11* mutations in the occurrence of myeloproliferative complications with the existence of high-risk mutations (*Mulero-Navarro et al., 2015*; *Strullu et al., 2014*). Given the aggressive nature of JMML and lack of therapies, a better understanding of the elusive JMML(-like MPN) pathophysiology and development of reliable preclinical models are essential.

Several lines of evidence indicate the importance of prenatal hematopoiesis such as the young age window for both NS-associated and sporadic JMML. The prenatal origin of *PTPN11* mutations in sporadic JMML suggests that JMML and NS-associated JMML-like MPN originate from fetal hematopoiesis (*Behnert et al., 2022*). Studying SHP2-driven JMML(-like MPN) during fetal hematopoiesis is challenging in conditional knock-in mouse models of JMML with *PTPN11* mutations, because mutant SHP2 expression is induced only postnatally (*Chan et al., 2009*; *Xu et al., 2011*) or indolent MPN is induced (*Tarnawsky et al., 2017*; *Tarnawsky et al., 2018*). NS knock-in mice with activating *PTPN11* mutations develop mild MPN only after 5 months of age (*Araki et al., 2004*; *Araki et al., 2009*).

Zebrafish (*Danio rerio*) with rapid ex utero development, transparent embryos, and conserved blood ontogeny emerged as a unique pediatric leukemia model that enables monitoring of leukemogenesis from its initial stages with high temporal and spatial resolution (*de Pater and Trompouki, 2018*; *Gore et al., 2018*). Furthermore, NS-associated features, such as shorter body axis length, craniofacial

defects, defective gastrulation, and impaired heart looping, are recapitulated in zebrafish embryos upon transient overexpression of NS-associated protein variants (*Bonetti et al., 2014*; *Jopling et al., 2007*; *Niihori et al., 2019*; *Paardekooper Overman et al., 2014*; *Runtuwene et al., 2011*).

To better assess the link between dysregulated SHP2 and myeloproliferation in the context of NS, we developed and characterized a novel genetic zebrafish model of NS with Shp2-D61G mutation, a dominantly inherited NS-associated mutation that is most frequently associated with NS/JMML-like MPN in human patients (*Strullu et al., 2014*; *Tartaglia et al., 2001*). Mutant zebrafish developed hematopoietic defects consistent with JMML-like MPN. Transcriptomic comparison of HSPCs obtained from JMML patients with somatic *PTPN11* mutations and HSPCs from mutant zebrafish harboring an NS-associated variant of Shp2 suggested common mechanisms of disease initiation in sporadic and syndromic *PTPN11*-driven JMML. Both datasets show a similar proinflammatory gene expression and an anti-inflammatory agent largely rescued the hematopoietic defects in mutant zebrafish, suggesting inflammation as potential drug target for sporadic and syndromic JMML(-like MPN). Our data show that Shp2 mutant zebrafish convincingly model the human NS-associated MPN thereby providing a valuable preclinical model for development of future therapies.

## Results
### Shp2$^{D61G}$ mutant zebrafish display typical NS traits

To investigate early hematopoietic defects associated with NS in a vertebrate animal model, we turned to zebrafish, which represents a versatile model to study leukemogenesis and in which it is feasible to introduce mutations at will using CRISPR/Cas9-mediated homology directed repair (*Tessadori et al., 2018*). NS is a dominant autosomal disorder and 50% of NS patients carry heterozygous mutations in the protein-tyrosine phosphatase SHP2 (*PTPN11*). The zebrafish genome contains two *ptpn11* genes (*ptpn11a* and *ptpn11b*), encoding Shp2a and Shp2b. Shp2a is indispensable during zebrafish development, whereas loss of Shp2b function does not affect development (*Bonetti et al., 2014*). Certain NS mutations are more frequently associated with NS/JMML-like MPN than others. The D61G substitution in SHP2 is the most frequently occurring (*Strullu et al., 2014*; *Tartaglia et al., 2001*). The homology between human SHP2 and zebrafish Shp2a is 91% and the amino acid D61 is located in an absolutely conserved region of the protein (*Figure 1—figure supplement 1A*). Thus, we introduced the D61G mutation in zebrafish Shp2a using CRISPR/Cas9-mediated homology directed repair (*Figure 1—figure supplement 1B*), targeting the *ptpn11a* gene. Sequencing confirmed that the oligonucleotide used for the homology directed repair was incorporated correctly into the genome (*Figure 1A*) and that the introduced mutation did not have a detectable effect on Shp2a protein expression (*Figure 1B*). Prior to phenotypic analyses, the mutant lines were outcrossed twice to ensure that potential background mutations due to the CRISPR/Cas9 approach were removed.

Genotypically, heterozygous Shp2$^{D61G/wt}$ zebrafish mutants correspond to NS patients. Consistently, in 32% of heterozygous Shp2$^{D61G/wt}$ and 45% of homozygous Shp2$^{D61G/D61G}$ embryos at 5 days post fertilization (dpf) we observed typical NS traits, such as reduced body axis extension, heart edema, and craniofacial deformities. Based on the extent of the phenotypes, defects were categorized as normal, mild, and severe (*Figure 1C and D*). The observed phenotypic defects were mostly mild, but in some cases severe defects were found, including severely stunted growth, edemas of the heart and jaw, and absence of the swim bladder (*Figure 1C and D*). A more detailed characterization of the typical NS traits showed that the body axis length was significantly reduced in Shp2$^{D61G}$ mutant embryos at 5 dpf (*Figure 1E*). Furthermore, imaging of Alcian blue-stained cartilage revealed NS-reminiscent craniofacial defects in 4 dpf Shp2$^{D61G}$ mutant embryos, characterized by broadening of the head (*Figure 1F*), leading to an increased ratio of the width of the ceratohyal and the distance to Meckel's cartilage (*Figure 1G*). We also assessed the general morphology and function of the mutant embryonic hearts. Whole-mount in situ hybridization (WISH) with cardiomyocyte (*myl7*), ventricular (*vhmc*), and atrial (*ahmc*) markers identified no obvious morphological heart defects, such as changes in heart size or heart looping, as well as heart chamber specification at 3 dpf (*Figure 1—figure supplement 1C*). A decrease in heart rate (*Figure 1I*), ejection fraction, and cardiac output (*Figure 1—figure supplement 1D,E*) was detected from the ventricular kymographs obtained from high-speed video recordings (*Tessadori et al., 2012*) of the Shp2$^{D61G}$ mutant hearts at 5 dpf (*Figure 1H*). Effects on cardiac function varied from embryo to embryo, which is reminiscent of variable heart defects in human patients.

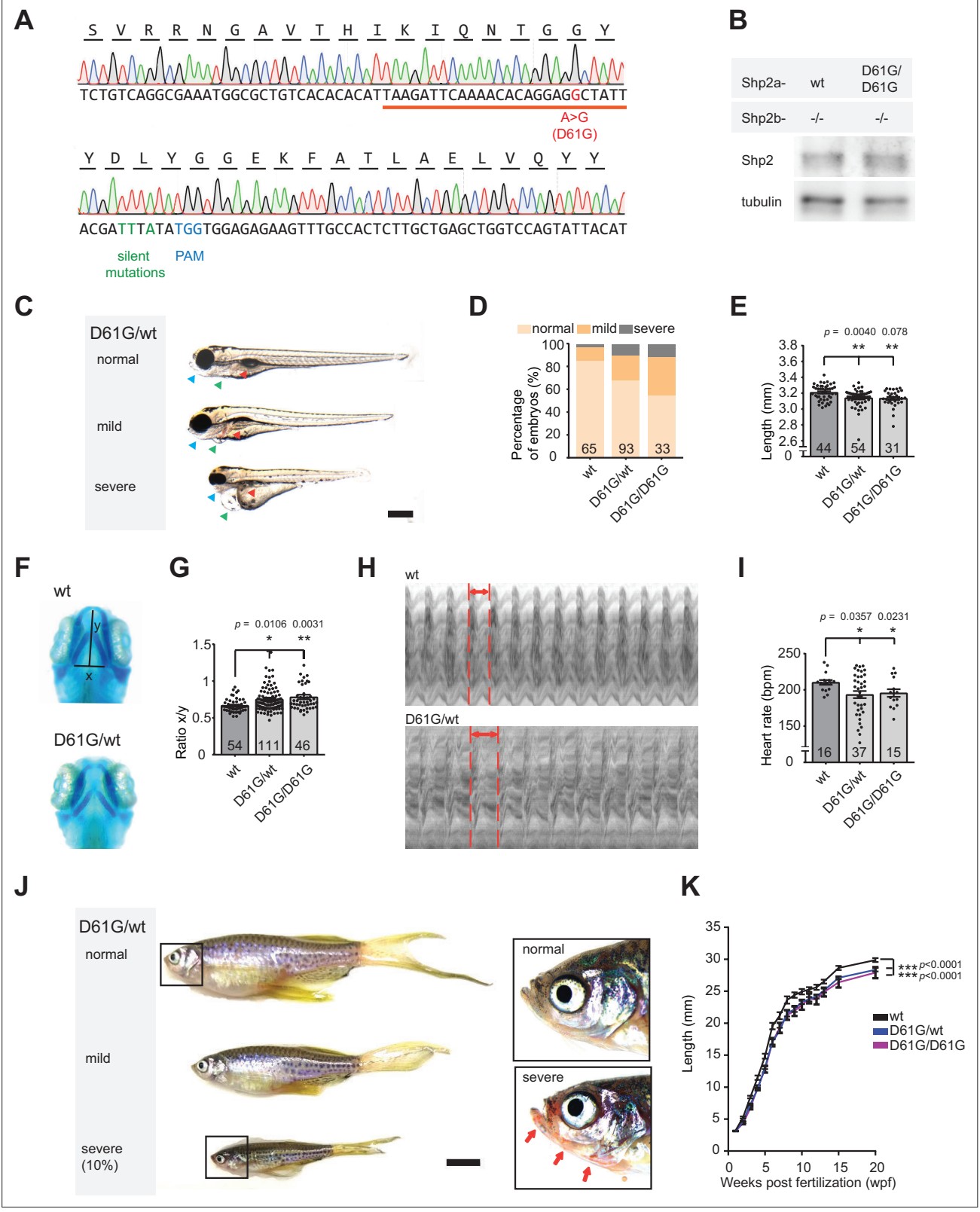

**Figure 1.** Shp2[D61G] zebrafish display Noonan syndrome (NS)-like traits. (**A**) Sequencing trace derived from Shp2[D61G/D61G] zebrafish. Oligonucleotide sequence used to generate the model is underlined. Nucleotide substitutions for D61G mutation (red), silent mutations close to the PAM site (green), and the PAM site (blue) are indicated. (**B**) Immunoblot of Shp2 levels from five pooled Shp2a[wt]Shp2b[-/-] or Shp2a[D61G/D61G]Shp2b[-/-] embryos using antibodies for Shp2 and tubulin (loading control) (**Figure 1—source data 1**). (**C**) Representative images of typical Shp2[D61G] zebrafish embryonic

*Figure 1 continued on next page*

*Figure 1 continued*

phenotypes at 5 days post fertilization (dpf). Blue arrows: jaw, green arrows: heart, red arrows: swim bladder. (**D**) Quantification of phenotypes of Shp2$^{wt}$, Shp2$^{D61G/wt}$, and Shp2$^{D61G/D61G}$ embryos scored as in (**C**) normal, mild, and severe. (**E**) Body axis length of Shp2$^{wt}$, Shp2$^{D61G/wt}$, and Shp2$^{D61G/D61G}$ embryos at 5 dpf. (**F**) Representative images of Alcian blue-stained head-cartilage of 4dpf Shp2$^{wt}$ and Shp2$^{D61G/wt}$ embryos (**x**) width of ceratohyal, (**y**) distance to Meckel's cartilage. (**G**) Quantified craniofacial defects (x/y ratio). (**H**) Representative ventricular kymographs derived from high-speed video recordings of beating hearts of 5 dpf Shp2$^{wt}$ and Shp2$^{D61G/wt}$ embryos. Red dotted lines indicate one heart period. (**I**) Heart rates derived from the ventricular kymographs. (**J**) Representative images of typical Shp2$^{D61G}$ zebrafish adult phenotypes at 24 weeks post fertilization (wpf). Red arrows indicate skin redness in the jaw region. Scale bar, 0.5 cm. Insets, zoom-in of boxed regions. (**K**) Body axis lengths of 10 Shp2$^{wt}$, 25 Shp2$^{D61G/wt}$, and 10 Shp2$^{D61G/D61G}$ zebrafish measured weekly between 5 dpf and 20 wpf of age. (**D,E,G,I**) Measurements originate from three distinct experiments. Number on bars: number of embryos. (**E,G,I,K**) Error bars: standard error of the mean (SEM), *p < 0.05; **p < 0.01, ***p < 0.001, ANOVA complemented by Tukey's HSD.

The online version of this article includes the following source data and figure supplement(s) for figure 1:

**Source data 1.** Immunoblot raw data showing uncropped and unedited blots of *Figure 1B*.

**Figure supplement 1.** Defective heart function, but not heart morphology in Shp2$^{D61G}$ zebrafish embryos.

The embryos with severe phenotypic defects did not develop a swim bladder and they did not survive to adulthood. The rest of both Shp2$^{D61G/wt}$ and Shp2$^{D61G/D61G}$ mutant zebrafish grew up normally, their life span was not affected, and they displayed rather mild defects in adult stages, such as shorter body axis (*Figure 1J*), with markedly reduced length observed from the embryonic stage of 5 dpf until the fully developed adults (*Figure 1K*). In 10% of Shp2$^{D61G}$ mutants the phenotypes were more severe, with markedly reduced body axis length compared to their siblings, skinny appearance and with overall redness, especially in the head and gill region (*Figure 1J*), which is not described in human NS patients and hence, may not be related to NS.

Taken together, the NS Shp2$^{D61G}$ mutant zebrafish we established phenocopied several of the typical NS traits, such as stunted growth, craniofacial defects, and heart defects. Similar to differences in symptoms observed in individual human patients, the severity of the defects varied among individual Shp2$^{D61G}$ mutant zebrafish.

## Increased numbers of HSPCs and myeloproliferative defects in Shp2$^{D61G}$ zebrafish embryos

Next, we explored hematopoietic abnormalities in the Shp2$^{D61G}$ zebrafish during embryonic development, corresponding to prenatal hematopoietic ontogeny in human. We were not able to observe any effect of the Shp2$^{D61G}$ mutation on primitive hematopoiesis in embryos at 2 dpf using WISH for markers of erythroid progenitors (*gata-1*), myeloid progenitors (*pu.1*), and white blood cells (*l-plastin*) (*Figure 2—figure supplement 1A*). On the other hand, a significant effect of the Shp2-D61G mutation on definitive hematopoiesis was observed at 5 dpf (*Figure 2*). First, the number of HSPCs was determined using the Tg(*cd41:GFP, kdrl:mCherry-CAAX*) transgenic line. CD41-GFP-positive cells mark HSPCs when the GFP signal is low (CD41-GFP$^{low}$) and thrombocytes when the GFP signal is high (CD41-GFP$^{high}$). An increased number of CD41-GFP$^{low}$ HSPCs was observed in the caudal hematopoietic tissue (CHT) region, corresponding to the fetal liver in human and the head kidney (corresponding to the bone marrow in human) of the mutant embryos (*Figure 2A and B*). Shp2$^{D61G}$ mutation seems to affect both proliferation and apoptosis of HSPCs, evident by an increase in CD41-GFP$^{low}$ cells positive for phosphohistone H3 (pHis3), a marker for late G2 and M phase and a decrease in Acridine orange-positive cells in the CHT region, marking apoptotic cells (*Figure 2—figure supplement 1B*,C). We observed an increase of *c-myb*-positive cells, marking HSPCs and of *l-plastin*-positive cells, marking all white blood cells, in both CHT and head kidney region of the Shp2$^{D61G}$ mutants (*Figure 2C–F*). These latter cells appear not to be lymphocytes, since the size of the *ikaros*-positive thymus was not affected (*Figure 2—figure supplement 1D*,E). On the other hand, the myeloid lineage was markedly expanded, evident from the increase in the number of mpx-GFP-positive neutrophils (*Figure 2G and H*) and mpeg-mCherry-positive macrophages (*Figure 2G1*), in Tg(*mpx:GFP, mpeg:mCherry*) double transgenic mutant embryos. Shp2$^{D61G}$ mutant embryos also displayed a mild decrease in the number of CD41-GFP$^{high}$-positive thrombocytes and the number of *β-globin*-positive erythrocytes (*Figure 2—figure supplement 1F-H*). The observed defects in all different blood lineages examined here were stronger in the homozygous Shp2$^{D61G/D61G}$ than heterozygous Shp2$^{D61G/wt}$ embryos, supporting a dosage effect. One of the clinical hallmarks of JMML is the hypersensitivity of myeloid progenitors to GM-CSF. GM-CSF is not present in zebrafish, but stimulation of HSPCs in ex vivo colony-forming

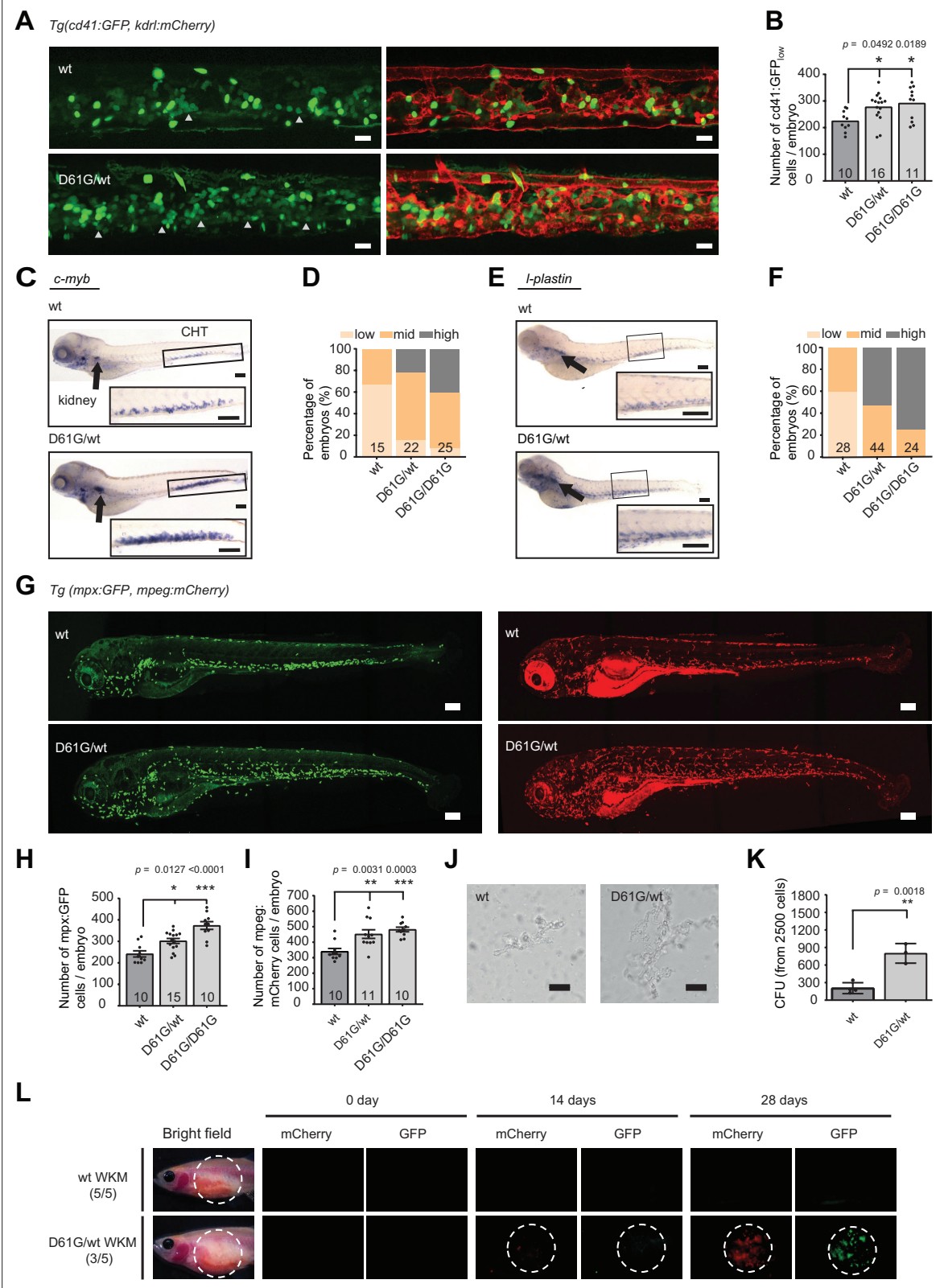

**Figure 2.** Shp2[D61G] mutant zebrafish embryos display juvenile myelomonocytic leukemia (JMML)-like myeloproliferative neoplasm (MPN). (**A**) Representative images of the caudal hematopoietic tissue (CHT) region of Shp2[wt] and Shp2[D61G] zebrafish embryos in Tg(*cd41:GFP, kdrl:mCherry-CAAX*) background at 5 days post fertilization (dpf). *cd41:GFP[low]* cells mark hematopoietic stem and progenitor cells (HSPCs) and *cd41:GFP[high]* cells thrombocytes. Gray arrow heads indicate *cd41:GFP[low]* cells. Scale bar, 20 μm. (**B**) The low-intensity *cd41:GFP*-positive cells in the CHT region were

*Figure 2 continued on next page*

Figure 2 continued

counted. (**C**) Whole-mount in situ hybridization (WISH) of 5 dpf Shp2$^{wt}$ and Shp2$^{D61G/wt}$ embryos using *c-myb*-specific probe. Head kidney (arrow) and CHT (box) are indicated; zoom-in in inset. Scale bars, 150 µm. (**D**) Quantification of *c-myb* WISH. *C-myb* expression in Shp2$^{wt}$, Shp2$^{D61G/wt}$, and Shp2$^{D61G/D61G}$ embryos was scored as low, mid, and high. (**E**) WISH of 5 dpf Shp2$^{wt}$ and Shp2$^{D61G/wt}$ embryos using *l-plastin*-specific probe. Head kidney (arrow) and CHT (box) are indicated and zoom-in in inset. Scale bars, 150 µm. (**F**) Quantification of *l-plastin* expression in Shp2$^{wt}$, Shp2$^{D61G/wt}$, and Shp2$^{D61G/D61G}$ embryos scored as low, mid, and high. (**G**) Representative images of Shp2$^{wt}$ and Shp2$^{D61G}$ zebrafish embryos in Tg(*mpx:GFP, mpeg:mCherry*) background at 5 dpf. *Mpx*:GFP marks neutrophils and *mpeg*:mCherry macrophages. Scale bars, 150 µm. (**H,I**) Number of *mpx*:GFP and *mpeg*:mCherry-positive cells per embryo. (**J**) Representative images of colonies developed from *cd41:GFP*$^{low}$ cells isolated from the CHT of 5 dpf Shp2$^{wt}$ and Shp2$^{D61G/wt}$ zebrafish embryos, grown in methylcellulose with zebrafish cytokine granulocyte colony stimulating factor a (Gcsfa) for 2 days. Scale bar, 50 µm. (**K**) Quantification of number of colonies from J, t-test. (**L**) WKM cells harvested from Shp2$^{wt}$ and Shp2$^{D61G}$ zebrafish in the Tg(*mpx:GFP, mpeg:mCherry*) background were injected into the peritoneum of adult *prkdc-/-* zebrafish. Recipients were monitored by fluorescence imaging. (**B,D,F,H,I,K**) Measurements originate from at least three distinct experiments. Number on bars: number of embryos. (**B,H,I,K**) Error bars represent SEM. *p < 0.05, **p < 0.01, ***p < 0.001. (**B,H,I**) ANOVA complemented by Tukey's HSD.

The online version of this article includes the following figure supplement(s) for figure 2:

**Figure supplement 1.** Hematopoiesis in Shp2$^{D61G}$ zebrafish.

assays in semi-solid media with granulocyte colony stimulating factor a (Gcsfa) or Gcsfb gives rise to colony-forming unit granulocyte (CFU-G) and colony-forming unit macrophage (CFU-M) type of colonies, similarly to GM-CSF stimulation in human, suggesting a compensatory role of Gcsfa/b for the missing GM-CSF in zebrafish (*Oltova et al., 2018*; *Stachura et al., 2013*). Compared to their wild type (wt) siblings, colonies developed from the CD41-GFP$^{low}$ cells isolated from the Shp2$^{D61G/wt}$ zebrafish embryos at 5 dpf and exposed to Gcsfa were larger in size and number, demonstrating an enhanced CFU-G and CFU-M colony-forming ability (*Figure 2J and K*). Finally, we tested whether the observed myeloid expansion was reconstituted upon transplantation of the WKM cells harvested from Shp2$^{D61G/wt}$ animals in the Tg(*mpx:GFP, mpeg:mCherry*) background into the optically clear recipient *prkdc-/-* immunodeficient zebrafish (*Moore et al., 2016*). Animals injected with 1 × 10$^5$ mutant WKM cells (3/5) accumulated GFP- and mCherry-positive cells near the site of injection starting at 14 days and increasing until 28 days. By contrast, animals injected with WKM cells from control sibling animals with wt Shp2a (5/5) lacked any GFP- and mCherry-positive cells (*Figure 2L*).

Our findings suggest that the Shp2$^{D61G}$ mutant zebrafish embryos develop multilineage hematopoietic defects during the definitive wave of fetal hematopoiesis, which originates in the HSPCs compartment. The observed defect is reminiscent of JMML-like MPN in human NS patients.

## Myeloid bias is established during early differentiation of Shp2$^{D61G}$ HSPCs

In an effort to better understand the pathogenesis mechanisms in the Shp2$^{D61G}$ mutant HSPCs, single-cell RNA sequencing was performed using SORT-Seq on CD41-GFP$^{low}$ cells derived from 5 dpf Shp2$^{wt}$, Shp2$^{D61G/wt}$, and Shp2$^{D61G/D61G}$ zebrafish embryos in the Tg(*cd41:GFP, kdrl:mCherry-CAAX*) transgenic background (*Figure 3A and B*). Unlike in human and mouse, the distinct compartments of HSPCs cannot be isolated, due to lack of antibodies and transgenic lines. Thus, to group cells based on their transcriptional program into specific HSPCs clusters, unsupervised clustering was performed using the RaceID3 package (*Herman and Grün, 2018*). Clusters were visualized by t-distributed stochastic neighbor embedding (t-SNE) and four major clusters were further analyzed (*Figure 3C*). Based on the differentially expressed genes (DEGs) and GO term analysis, cells from Cluster 1 were determined to be thrombocyte and erythroid progenitors, cluster 2 hematopoietic stem cell (HSC)-like HSPCs, cluster 3 early myeloid progenitors, and cluster 4 monocyte/macrophage progenitors (*Figure 3C*, *Figure 3—figure supplement 1A*, *Supplementary file 1*). A small subset of cells in cluster 4 represented more differentiated neutrophil progenitors (*Figure 3—figure supplement 1B*).

HSPCs of either Shp2$^{wt}$ or mutant Shp2$^{D61G/wt}$ and Shp2$^{D61G/D61G}$ genotypes were present in all four major clusters, indicating that distinct HSPCs phenotypes were maintained on a gene transcription level. However, the distribution of cells in clusters differed among genotypes. An overrepresentation of mutant Shp2$^{D61G/wt}$ and Shp2$^{D61G/D61G}$ cells was observed in the HSC-like HSPCs cluster and monocytes/macrophages progenitors cluster, whereas these were underrepresented in the thrombocyte and erythroid progenitors cluster (*Figure 3D and E*). To validate this observation, we investigated the expression of *pu.1* and *alas2* markers in Shp2$^{D61G}$ embryos of different genotypes by WISH. In the

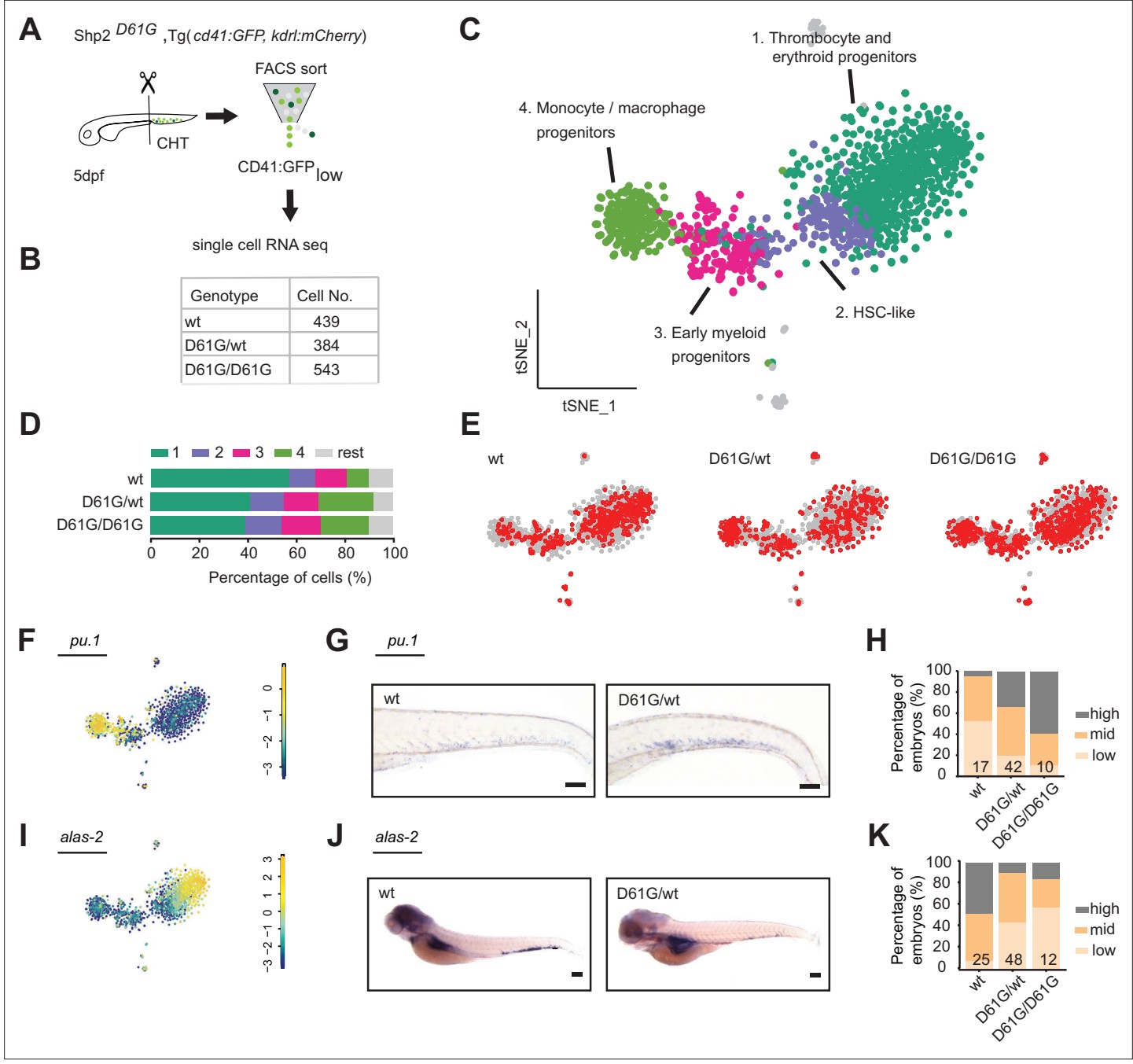

**Figure 3.** Single-cell RNA sequencing of hematopoietic stem and progenitor cells (HSPCs) reveals myeloid bias in Shp2$^{D61G}$ embryos. (**A**) Schematic representation of the experimental procedure. At 5 days post fertilization (dpf), caudal hematopoietic tissues (CHTs) from Shp2$^{wt}$, Shp2$^{D61G/wt}$, and Shp2$^{D61G/D61G}$ embryos in Tg(*cd41:GFP, kdrl:mCherry-CAAX*) background were isolated. Cells were dissociated and separated by fluorescence-activated cell sorting (FACS), based on *cd41:GFP$^{low}$* expression, prior to single-cell RNA sequencing, as described in the Materials and methods section. (**B**) Number of cells of distinct genotypes used in single-cell RNA sequencing analysis. (**C**) Combined t-distributed stochastic neighbor embedding (t-SNE) map generated using the cells of all three genotypes (Shp2$^{wt}$, Shp2$^{D61G/wt}$, and Shp2$^{D61G/D61G}$). Single cells from four major clusters are marked in dark green, blue, pink, and green, and their identities based on marker gene expression are indicated. Minor clusters are marked in gray. (**D**) Barplots showing the percentage of cells of Shp2$^{wt}$, Shp2$^{D61G/wt}$, and Shp2$^{D61G/D61G}$ genotype in distinct clusters. (**E**) Cells of distinct genotypes Shp2$^{wt}$, Shp2$^{D61G/wt}$, and Shp2$^{D61G/D61G}$ are visualized in t-distributed stochastic neighbor embedding (t-SNE) maps in red. (**F**) t-SNE maps showing log2-transformed read-counts of *pu.1*. (**G**) Representative images of the WISH staining for *pu.1* expression in 5 dpf Shp2$^{wt}$ and Shp2$^{D61G}$ zebrafish embryos. Scale bar, 100 μm. (**H**) Expression of the *pu.1* marker scored as low, mid, and high. (**I**) t-SNE maps showing log2-transformed read-counts of *alas-2*. (**J**) Representative images of the WISH staining for *alas-2* expression in the tail region of 5 dpf Shp2$^{wt}$ and Shp2$^{D61G}$ zebrafish embryos. Scale bar, 100 μm. (**K**) Expression of the *alas-2* marker scored as low, mid, and high. (**H,K**) Number on bars: number of embryos.

*Figure 3 continued on next page*

*Figure 3 continued*

The online version of this article includes the following figure supplement(s) for figure 3:

**Figure supplement 1.** Identification of different hematopoietic stem and progenitor cells (HSPCs) subpopulations based on differential expression of representative genes.

single-cell RNA sequencing dataset, expression of *pu.1* and *alas2* was upregulated in the myeloid progenitors and erythroid progenitors, respectively (*Figure 3F1*). An increased number of *pu.1*-positive cells was detected by WISH in 5 dpf old Shp2$^{D61G}$ embryos compared to their Shp2$^{wt}$ siblings (*Figure 3G and H*), whereas the number of *alas2*-positive cells in Shp2$^{D61G}$ mutants was decreased (*Figure 3J and K*).

Taken together, single-cell RNA sequencing suggests that defects during early differentiation of HSPCs to monocyte/macrophage progenitors and erythroid/thrombocyte progenitors initiate the multilineage blood defect observed in Shp2$^{D61G}$ embryos, which recapitulates features of NS-JMML-like MPN.

## Excessive proinflammatory response in monocyte/macrophage progenitors of Shp2$^{D61G}$ HSPCs

We further analyzed the cluster of monocyte/macrophage progenitors, in which we observed an over-representation of mutant Shp2$^{D61G}$ cells (*Figure 4A*). Functional annotation of the DEGs and the GO term enrichment analysis revealed enhanced expression of proinflammatory genes in mutant Shp2$^{D61G}$ cells (*Figure 4A and B*, *Supplementary file 1*). Interestingly, expression of inflammation-related genes, such as *gcsfa*, *gcsfb*, *il1b*, *irg1,* and *nfkbiaa,* was constrained to the cells of mutant Shp2$^{D61G}$ genotype in the monocyte/macrophage progenitors cluster, whereas the monocyte marker *timp2b* was equally expressed by cells of distinct genotypes (*Figure 4B*).

We next aimed to determine at which point the observed inflammatory program initiated during the differentiation of HSC-like cells into monocyte/macrophage progenitors. We used Monocle trajectory inference to order cells from the four major clusters based on their differentiation state along the pseudotime. HSC-like cells were located at the start of the progression trajectory and from there, two differentiation routes were determined, one leading to the thrombocyte and erythrocyte progenitors, and the second one leading to the monocyte/macrophage progenitors (*Figure 4C and D*). The route that includes HSC-like cells, early myeloid progenitors, and monocyte/macrophage progenitors was then selected (*Figure 4C and D*, *Figure 4—figure supplement 1A*) and the expression of inflammatory genes in distinct genotypes was computed along it (*Figure 4E*, *Figure 4—figure supplement 1B*). Expression of the monocyte marker *timp2b* started during early differentiation of monocyte/macrophage progenitors. The level of *timp2b* expression was similar in Shp2$^{wt}$ and Shp2$^{D61G}$ HSPCs. Expression of the inflammatory genes, *il1b*, *gcsfa*, *irg1*, *nfkbiaa,* and *tnfa*, started at a similar pseudotime as *timp2b* expression, but with a more robust expression in Shp2$^{D61G}$ HSPCs than in Shp2$^{wt}$ HSPCs. Finally, high expression of IL-1β was validated in mutant Tg(*il1b:eGFP*, *mpeg:mCherry*) transgenic embryos in vivo. IL-1β-eGFP-positive cells were more abundant in the CHT region of Shp2$^{D61G}$ embryos than in wt embryos (*Figure 4F*).

Taken together, Shp2$^{D61G}$ embryos display proinflammatory gene expression in HSPCs, which is initiated during early differentiation of the monocyte/macrophage progenitor cells.

## *PTPN11* somatic mutations induce inflammatory response genes in HSPCs of JMML patients

We hypothesized that activating mutations in Shp2 may trigger an inflammatory response in a similar way in sporadic JMML as we observed in syndromic JMML-like MPN. To investigate this, we performed transcriptomic analysis of the HSPCs compartments derived from bone marrows of sporadic JMML patients with activating mutations in SHP2 (n = 5) and age-matched healthy donors (n = 7). The SHP2 mutations in JMML HSPCs were distinct among patients and both HSCs and progenitors were included (*Supplementary file 2*). Overall, we identified 1478 DEGs in HSPCs from JMML patients, compared to the healthy donor HSPCs (*Figure 5A*, *Supplementary file 3*). The functional analysis of DEGs using gene set enrichment analysis (GSEA) was performed to systematically explore hallmark gene signatures specific for JMML HSPCs (*Figure 5B*, *Supplementary file 4*). Strikingly, altered

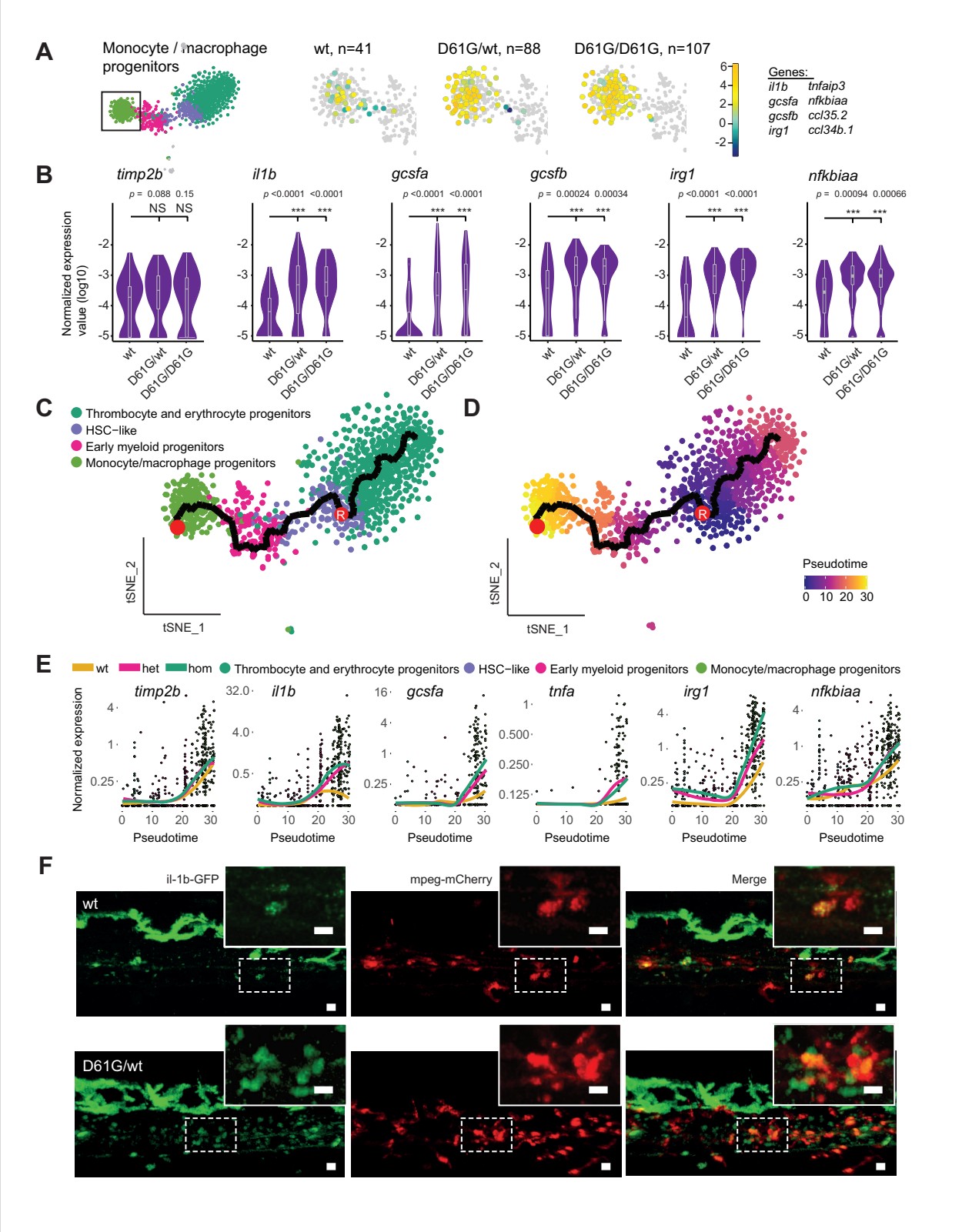

**Figure 4.** Inflammatory response in monocyte/macrophage progenitors in Shp2[D61G] embryos. (**A**) The monocyte/macrophage progenitors cluster (boxed on the t-distributed stochastic neighbor embedding [t-SNE] map on left) was analyzed in detail. log2-transformed sum of read-counts of selected inflammation-related genes from the top 50 differentially expressed genes in the cells of the monocyte/macrophage progenitors cluster, with genotype and number of cells (**n**) indicated above. (**B**) Violin plots show the expression of specific genes in monocyte/macrophage progenitors of Shp2[wt], Shp2[D61G/]

*Figure 4 continued on next page*

*Figure 4 continued*

*wt*, and Shp2*D61G/D61G* genotypes. NS, not significant, \*\*\*p < 0.001, t-test. (**C,D**) Monocle pseudotime trajectory of Shp2*wt*, Shp2*D61G/wt*, and Shp2*D61G/D61G* hematopoietic stem and progenitor cells (HSPCs) differentiation superimposed on the four clusters and indicated by color in D. A route leading from hematopoietic stem cell (HSC)-like cells to monocyte/macrophage progenitors was selected and its root (**R**) and end are marked with red circles. (**E**) Pseudotemporal expression dynamics of specific genes along the selected route. The lines represent the smoothened regression of the moving average for Shp2*wt*, Shp2*D61G/wt*, and Shp2*D61G/D61G* genotypes. (**F**) In vivo imaging of the caudal hematopoietic tissue (CHT) region of Shp2*wt* and Shp2*D61G/wt* zebrafish embryos in Tg(*il1b:eGFP, mpeg:mCherry*) background at 5 days post fertilization (dpf). Representative images are shown. The dashed line boxed region of CHT is zoom-in in inset. Scale bar, 10 μm.

The online version of this article includes the following figure supplement(s) for figure 4:

**Figure supplement 1.** Distinct expression profiles of mutant zebrafish embryos in pseudotime.

expression of genes related to inflammation was the most prominent, such as genes involved in TNFα signaling via NFκB. The inflammatory response genes were significantly enriched in JMML HSPCs (*Figure 5B–D*). De-regulation of genes related to proliferation (G2M checkpoint, MYC targets, and E2F targets) was evident in JMML HSPCs, suggesting their decreased quiescence (*Figure 5B*).

We next studied whether the inflammatory-related gene signature overlaps between HSPCs from zebrafish NS/JMML-like MPN model and human JMML. GSEA revealed that human orthologs of the top 100 DEGs found in the zebrafish monocyte/macrophage progenitor cells were enriched significantly in patient JMML HSPCs (*Figure 5E*). Some of the top overexpressed inflammation-associated genes in JMML patients, such as TNF and MARCKSL, were also overexpressed in NS/JMML-like MPN mutant zebrafish embryos compared to the wt (*Figure 5F and G*). 3D principal component analysis (PCA) visualized that these zebrafish signature genes were able to clearly segregate HSPCs from JMML patients and healthy donors (*Figure 5H*). These findings suggest that mutant, activated SHP2 triggers proinflammatory gene expression in HSPCs both in sporadic JMML patients and in our zebrafish model of syndromic JMML-like MPN in a similar manner, suggesting a common underlying, endogenously driven process.

## Inhibition of the proinflammatory response ameliorates the JMML-like MPN phenotype

To investigate the role of the inflammatory response in the blood defect, we assessed the effect of dexamethasone in Shp2*D61G* embryos. Dexamethasone is an anti-inflammatory agent, which is known to suppress the inflammatory response in the monocyte/macrophage lineage (*Ehrchen et al., 2019*; *Xie et al., 2020*). In parallel, inhibitors of the known Shp2-associated signaling pathways were used, targeting MEK (CI1040) or PI3K (LY294002) (*Tajan et al., 2015*). Shp2*D61G* embryos were exposed to inhibitors continuously from 2 to 5 dpf (*Figure 6A*). Both CI1040 and LY294002 led to a robust reduction in expansion of both HSPCs and myeloid lineage in mutant embryos, measured by WISH using markers for *c-myb* and *l-plastin* (*Figure 6B and C*), demonstrating that the NS-associated blood development defect in Shp2*D61G* embryos was largely caused by overactivation of the RAS-MAPK and PI3K pathways as expected.

To our surprise, dexamethasone reduced *c-myb* and *l-plastin* expression to a similar extent as CI1040 and LY294002 (*Figure 6B and C*), indicating a profound role for the inflammatory response in the pathogenesis of NS-associated blood defects. We then examined whether dexamethasone treatment affected the expanded myeloid lineage in Shp2*D61G* embryos by determining the number of neutrophils and macrophages, in the Tg(*mpx:GFP, mpeg:mCherry*) double transgenic mutant embryos (*Figure 6D–F*). Upon dexamethasone treatment, the number of *mpx*-GFP and *mpeg*-mCherry-positive cells decreased in Shp2*D61G* embryos to similar levels as observed in Shp2*wt* embryos, indicating that dexamethasone treatment rescued the expansion of the myeloid lineage in Shp2*D61G* embryos. Dexamethasone did not significantly alter the number of neutrophils or macrophages in wt embryos (*Figure 6D–F*). Finally, the expression of proinflammatory genes, such as *tnfa*, *il1b*, and *gcsfb* in Shp2*D61G* embryos was reduced in response to dexamethasone treatment to levels observed in Shp2*wt* embryos (*Figure 6G*).

Taken together, we observed a proinflammatory phenotype of the macrophage/monocyte lineage in Shp2*D61G* mutant zebrafish, which was established early during HSPCs differentiation. The anti-inflammatory agent dexamethasone largely rescued the hematopoietic defects, suggesting that the

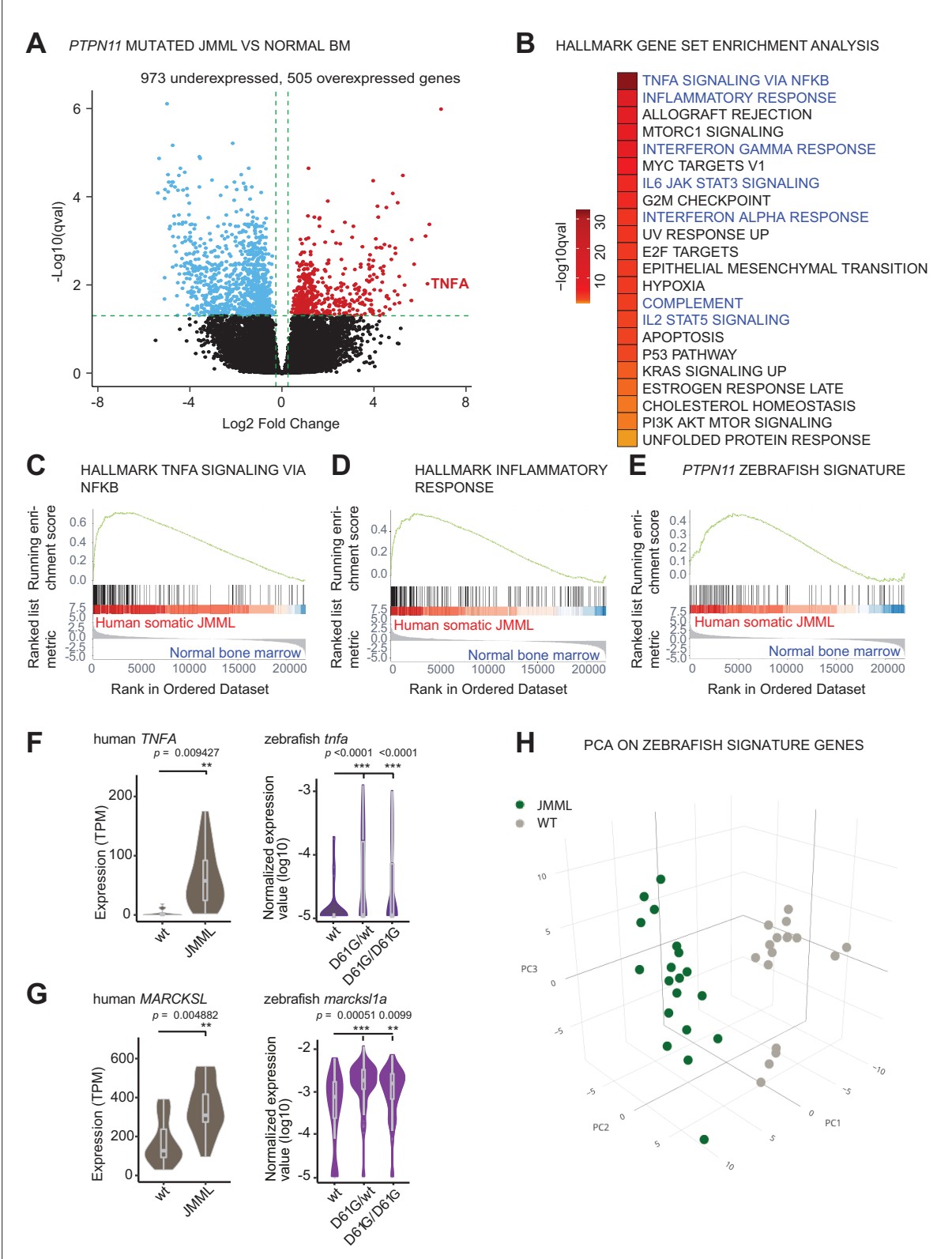

**Figure 5.** Similar molecular signatures in hematopoietic stem and progenitor cells (HSPCs) from human juvenile myelomonocytic leukemia (JMML) patients and zebrafish Shp2$^{D61G}$ embryos. (**A**) Volcano plot of differentially expressed genes of HSPCs derived from bone marrow of JMML patients with *PTPN11* mutations (n = 5) and healthy bone marrow (n = 7). Underexpressed genes are marked in blue and overexpressed genes are marked in red. *TNFA* expression is highlighted. Green dashed lines indicate the significance level. (**B**) Gene set enrichment analysis (GSEA) for the MSigDB's

*Figure 5 continued on next page*

*Figure 5 continued*

hallmark gene sets in HSPCs from JMML compared to normal human age-matched bone marrow. GSEA plots for TNFA_SIGNALING_VIA_NFKB (**C**), INFLAMMATORY RESPONSE (**D**), and the custom zebrafish signature based on the top 100 human orthologous of genes upregulated in monocyte/macrophage progenitor cluster of zebrafish HSPCs (**E**). Violin plots show the expression of *TNFA* (**F**) and *MARCKSL* (**G**) in either human wild type (wt) vs. JMML HSPCs, and zebrafish Shp2$^{wt}$ vs. Shp2$^{D61G/wt}$ and Shp2$^{D61G/D61G}$ monocyte/macrophage HSPC progenitors. **p < 0.01, ***p < 0.001, t-test. (**H**) Principal component analysis (PCA) for the 100 genes included in the custom zebrafish signature. Green dots represent *PTPN11* mutated JMML, gray ones represent normal human age-matched bone marrow. PC1: 19% of the variance, PC2: 16% of the variance, PC3: 14% of the variance.

inflammatory response in Shp2$^{D61G}$ mutant zebrafish had a causal role in the pathogenesis of the NS/JMML-like MPN blood phenotype.

## Discussion

To investigate the timing and pathophysiology of mutant SHP2-related myeloproliferative defects, we explored the transcriptomes of HSPCs with activating mutations in SHP2, derived from sporadic JMML patients and syndromic NS/JMML-like MPN zebrafish embryos, respectively. Proinflammatory gene expression was evident both in HSPCs from JMML patients and in mutant zebrafish. It should be noted that the type of data is different, bulk sequencing of the human samples and single-cell sequencing of the zebrafish material. In addition, the source of the material is distinct, fluorescence-activated cell sorting (FACS)-sorted CD41-GFP$^{low}$ HSPCs from zebrafish embryos and immunophenotypically FACS-sorted postnatal HSCs and progenitors from human. Moreover, genotypically, the samples differed in that the zebrafish carried a single Shp2 mutation (D61G) and the human JMMLs carried distinct activating SHP2 mutations even with additional mutations. Therefore, it is not surprising that the ranking of overexpressed inflammatory genes in HSPCs from mutant zebrafish and human JMML patients was not identical. Nevertheless, the similarity of the proinflammatory signatures revealed by GSEA between these diverging samples is striking.

Already early studies reported high levels of proinflammatory cytokines, such as IL-1β, TNF-α, GM-CSF, in JMML patients plasma (**Bagby et al., 1988**; **Freedman et al., 1992**). However, little is known about the cellular origin of the proinflammatory status, its pathophysiological role, and therapeutic potential. The inflammatory response may be initiated in a cell autonomous way or in response to signals from the microenvironment. Dong et al. suggested that IL-1β is secreted by differentiated monocytes which get recruited upon secretion of the chemokine, CCL3, by cells of the bone marrow microenvironment containing activating SHP2 mutations. However, the levels of CCL3 in the bone marrow of four NS patients varied (**Dong et al., 2016**). Furthermore, a microenvironment-driven inflammatory response would not explain high levels of cytokines in sporadic JMML, where niche cells are not mutated. On the other hand, recent reports indicate that JMML leukemia stem cells are heterogeneous but confined to the HSPCs compartment, defining JMML HSPCs as the origin of the disease (**Caye et al., 2020**; **Louka et al., 2021**). The pseudotime analysis of inflammatory genes in zebrafish HSPCs indicated that the initiation of the proinflammatory program happens during early differentiation of HSPCs toward the monocyte/macrophage progenitor lineage in Shp2$^{D61G}$ embryos. The limited number of human patients included in our analysis allowed us to establish that the HSPCs compartment from JMML patients showed an inflammatory response, but not which lineage. Transcriptomic analysis of individual progenitor lineages from additional patients will be required to conclude which cell type displayed an inflammatory response. Our data suggest that proinflammatory reprogramming of the monocyte/macrophage lineage might be endogenously driven at least in part, and detailed mechanisms remain to be elucidated in the future.

To our knowledge, the Shp2$^{D61G}$ mutant zebrafish we developed is the first NS zebrafish model carrying an activating Shp2 mutation at its endogenous locus generated by CRISPR/Cas9-based knock-in technology. Strikingly, phenotypes developed by the Shp2$^{D61G}$ zebrafish corroborate closely with the phenotypes displayed by human NS patients and the existing NS mouse models (**Figure 1**). Our model presents an exciting novel tool for deciphering pathogenesis mechanisms of NS with its complex traits and for finding novel therapies for this as yet poorly treatable condition.

NS children with D61G mutation have a higher predisposition to develop JMML(-like MPN), and comparably Shp2$^{D61G}$ zebrafish embryos displayed typical JMML-like MPN characteristics, such as expansion of the myeloid lineage, increased sensitivity to Gcsfa, mild anemia, and thrombocytopenia

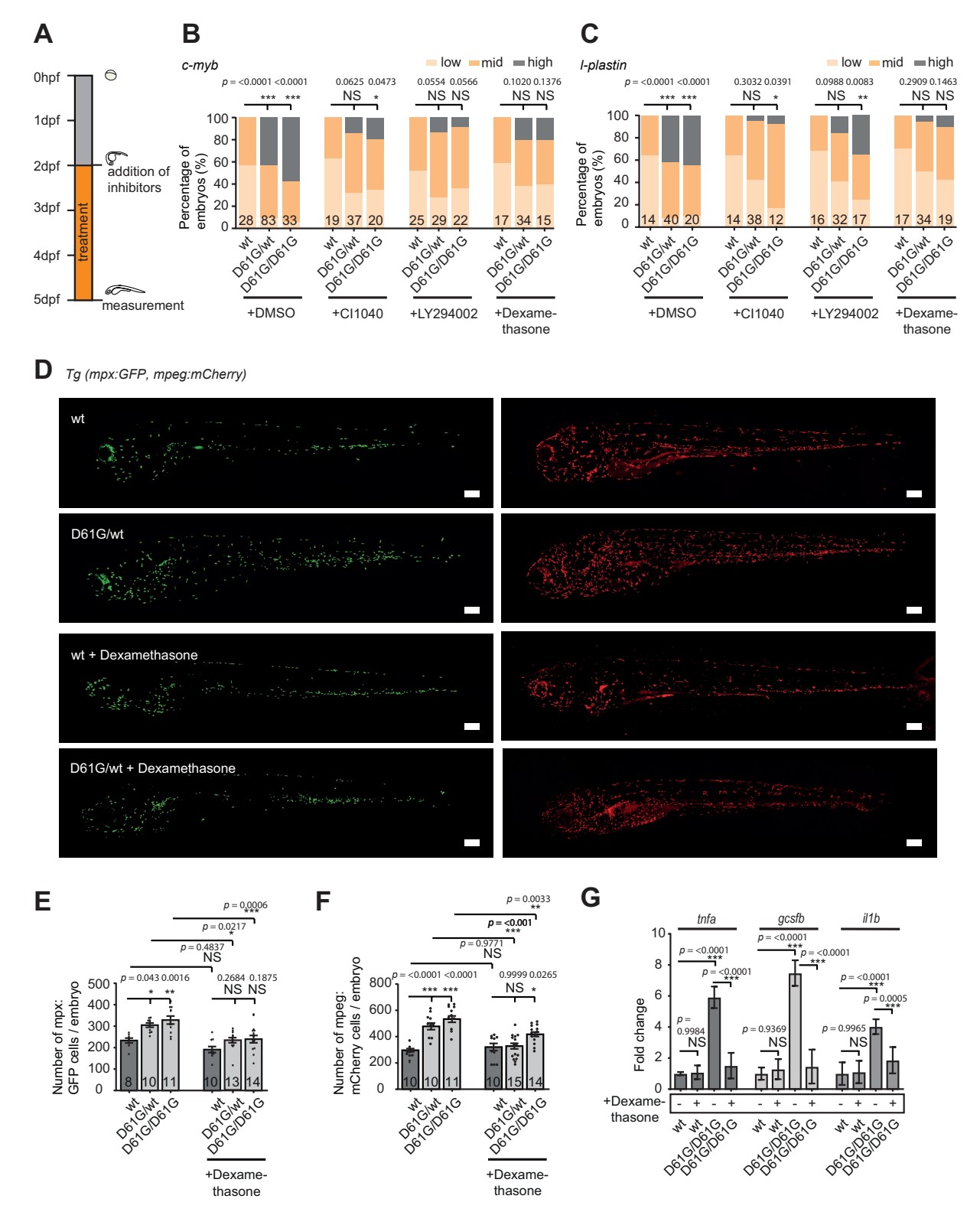

**Figure 6.** Anti-inflammatory treatment of zebrafish Shp2$^{D61G}$ embryos ameliorates the juvenile myelomonocytic leukemia (JMML)-like myeloproliferative neoplasm (MPN) phenotype. (**A**) Schematic overview of the treatments with MEK inhibitor CI1040, PI3K inhibitor LY294002, and anti-inflammatory corticosteroid dexamethasone. Embryos were continuously treated from 48 hr post fertilization (hpf) until 5 days post fertilization (dpf), when either whole-mount in situ hybridization (WISH), confocal imaging, or quantitative reverse transcription PCR (RT-qPCR) was performed. (**B,C**) WISH staining for

*Figure 6 continued on next page*

*Figure 6 continued*

the expression of the *c-myb* and *l-plastin* markers was scored as low, mid, and high. Measurements originate from at least three distinct experiments. Number on bars: number of embryos. NS, not significant; *p < 0.05, **p < 0.01, ***p < 0.001, chi-squared test. (**D**) Representative images of 5 dpf Shp2$^{wt}$ and Shp2$^{D61G}$ zebrafish embryos in Tg(*mpx:GFP, mpeg:mCherry*) background without and with dexamethasone treatment. Scale bars, 150 µm. (**E,F**) Number of *mpx*:GFP and *mpeg*:mCherry-positive cells per embryo. Number on bars: number of embryos. Error bars represent SEM. (**G**) Expression of *tnfa*, *gcsfb* and *il1b* genes determined by RT-qPCR in Shp2$^{wt}$ and Shp2$^{D61G/D61G}$ zebrafish embryos without and with dexamethasone treatment at 5 dpf, normalized to *ef1a* expression. Standard deviation of four samples in duplicates for each condition are shown. (**E,F,G**) *p < 0.05, **p < 0.01, ***p < 0.001. ANOVA complemented by Tukey's HSD.

(*Figure 2*). Furthermore, the observed blood defect was transplantable to secondary recipients and the disease originated in HSPCs, which displayed aberrant proliferation and apoptosis. Given the importance of fetal hematopoiesis during JMML(-like MPN) development, the zebrafish Shp2$^{D61G}$ mutant represents a reliable and unique model for JMML-like MPN, which allows us to study hematopoietic defects caused by mutant Shp2 during the prenatal development for the first time. Our results suggest that the JMML-like MPN defect is initiated at the CHT, which is the counterpart of fetal liver in human. Here, we observed expanded HSPCs, which displayed aberrant proliferation and apoptosis. We further characterized the transcriptomes of HSPCs specifically originating from the CHT, and studied their response to inhibitor treatments directly at the CHT. Striking similarities between the patient and zebrafish HSPCs transcriptomes indicate that the Shp2$^{D61G}$ zebrafish model is an exciting preclinical model for in vivo drug screens at relevant developmental time points, in a high throughput manner.

The only effective treatment of JMML is allogeneic stem cell transplantation, which has a high relapse rate of 50%. Hence, there is a great need for other means of therapeutic intervention. The role of inflammation as one of the drivers in myeloid leukemogenesis is emerging (*Arranz et al., 2017*; *Craver et al., 2018*). Here, we demonstrate that dampening inflammation using the glucocorticoid dexamethasone partially rescued the observed blood phenotype in Shp2$^{D61G}$, suggesting that the inflammatory response evoked in the cells of myeloid/macrophage lineage was an important driver of the NS/JMML-like blood defect and might be a potential drug target for both sporadic and syndromic JMML(-like MPN). To our surprise, targeting inflammation reversed the blood defect to a similar extent as the MAPK and PI3K pathway inhibitors, emphasizing not only the crucial role of inflammation during JMML pathogenesis, but also its strong therapeutic potential. Since anti-inflammatory therapies are non-invasive and widely available, targeting inflammation might represent a novel avenue for JMML treatment, either alone or in combination with other therapies.

In conclusion, we observed striking similarities in expression patterns of HSPCs from sporadic human JMML patients with an activating SHP2 mutation and HSPCs from an engineered zebrafish model with an activating NS-associated mutation in Shp2. Particularly genes associated with the inflammatory response were upregulated and strikingly, pharmacological inhibition of the inflammatory response ameliorated JMML-like MPN in the zebrafish model, suggesting this may be a first step for therapeutic intervention in human patients.

# Materials and methods
## Zebrafish husbandry

All procedures involving experimental animals were approved by the animal experiments committee of the Royal Netherlands Academy of Arts and Sciences (KNAW), Dierexperimenten commissie protocol HI18-0702, and performed under the local guidelines in compliance with national and European law. The following zebrafish lines were used in the study: Tübingen longfin (TL, wt), Tg(*cd41:GFP, kdrl:mCherry-CAAX*) (*Hogan et al., 2009*; *Lin et al., 2005*), Tg(*mpx:GFP, mpeg:mCherry*) (*Ellett et al., 2011*; *Renshaw et al., 2006*), TgBAC(*il1b:eGFP*)sh445 (*Ogryzko et al., 2019*), ptpn11b-/- (*Bonetti et al., 2014*), prkdc-/- (*Moore et al., 2016*), and the novel Shp2$^{D61G}$ zebrafish line. Raising and maintenance of zebrafish was performed according to *Aleström et al., 2020*; *Westerfield, 2000*. When required, pigmentation of embryos was blocked by adding phenylthiourea (PTU) (Sigma-Aldrich, St Louis, MO, Ref: P7629) at a concentration of 0.003% (v/v) to the E3 medium at 24 hr post fertilization (hpf).

## Patient material

All children's samples were obtained after parents had given their written informed consent. Experiments were approved by the institutional review board of the French Institute of Health and Medical Research (INSERM) (IORG0003254) in accordance with the Helsinki declaration. Healthy children bone marrows were obtained from intrafamilial BM transplantation donors and used with the approval of the Institutional Review Board of 'Hôpitaux Universitaires Paris Nord Val-de-Seine', Paris 7 University, AP-HP (IRB: 00006477), in accordance with the Helsinki declaration.

HSPCs fractions were FACS-sorted according to immunophenotypic signature as previously described (*Caye et al., 2020*), from PTPN11$^{mut}$ JMML (n = 5) and healthy children bone marrow (n = 7). Age, sex, mutations, and fractions available for each sample (HSCs, multipotent progenitors, common myeloid progenitors, granulocyte-macrophage progenitors, and megakaryocyte-erythroid progenitors) are indicated in *Supplementary file 2*.

## Generation of the Shp2$^{D61G}$ zebrafish line

The Shp2$^{D61G}$ zebrafish line was generated using the previously described CRISPR/Cas9-based knock-in approach (*Tessadori et al., 2018*). The sgRNA targeting exon 3 of the *ptpn11a* gene (5'-GGAGACTA TTACGACCTGTA-3') was designed using the CHOP-CHOP database (http://chopchop.cbu.uib.no/), further processed according to the previously published guidelines (*Gagnon et al., 2014*) and finally transcribed using the Ambion MEGAscript T7 kit (Thermo Fisher Scientific, Waltham, MA, Ref: AMB13345). The sgRNA, constant oligonucleotide, and template oligonucleotide were all generated by Integrated DNA Technologies (Coralville, IA) as standard desalted oligos and template oligonucleotide was further purified using the QIAquick Nucleotide Removal Kit (Qiagen, Hilden, Germany, Ref: 28304). The oligonucleotide used for the homology repair is 59 nucleotides long and besides the D61G mutation, contains three additional silent mutations in proximity to the PAM site. The sequence of template oligonucleotide is 5'-GAGTGGCAAACTTCTCTCCACCATATAAATCGTAATAGCCTCCT GTGTTTTGAATCTTA-3'. TL wt zebrafish embryos at the one-cell stage were injected directly in the cell with 1 nl of the injection mixture containing 18.75 ng/ml of sgRNA, 37.5 ng/ml of template oligonucleotide, and 3.6 mg/ml of Cas9 protein in 300 mM KCl. Cas9 protein was a gift from the Niels Geijsen laboratory at the Hubrecht Institute. The injected embryos were grown into the adulthood. F0 generation zebrafish were outcrossed with the wt zebrafish. DNA extracted from the 12 distinct 1 dpf old F1 generation embryos was screened for the correct insertion of the template oligonucleotide. Screening was done by Sanger sequencing (Macrogen Europe B.V., Amsterdam, The Netherlands) of the 225 bp long PCR product encompassing the genomic regions of the CRISPR target sites, which was generated using the forward 5'-TCATCTCCTCACTAGGCGAAAT-3' and reverse primer 5'-TATGTATGTGCTCACCTCTCGG-3'. The efficiency of the knock-in was 1.8% (1 founder zebrafish in 54 screened primary injected zebrafish). F1 generation was then established from the F0 founder. F1 generation adults were finclipped and sequenced for the presence of the mutation. All experiments were performed in zebrafish embryos and adults from the F3 and F4 generation. For all of the experiments except single-cell RNA sequencing and quantitative reverse transcription PCR (RT-qPCR), embryos were siblings derived from an incross of Shp2$^{D61G/wt}$ animals. After the experimental procedure, embryos were lysed and genotyped by sequencing as described above. Western blotting was performed as previously described (*Hale and den Hertog, 2018*) using the Shp2 (Santa Cruz Biotechnology, Dallas, TX, Ref: SC-280) and β-tubulin (Merck Millipore, Burlington, MA, Ref: CP06) antibodies.

## Phenotyping of the NS traits

Body axis lengths were measured from the tip of the head to the end of the trunk in the bright-field images of laterally positioned embryos, larva, and adults, which were anesthetized in 0.1% MS-222. Alcian blue (Sigma-Aldrich, Ref: A5268) staining was performed as previously described (*Paardekooper Overman et al., 2014*), on PTU-treated 4 dpf old embryos, which were anesthetized in 0.1% MS-222 and fixed in 4% PFA overnight. Embryos were positioned on their back in 70% glycerol in PBS and imaged with Leica M165 FC stereomicroscope (Leica Microsystems, Wetzlar, Germany). Analysis was performed in ImageJ. In vivo high-speed bright-field imaging of the embryonic hearts from PTU-treated embryos at 5 dpf, which were anesthetized in 0.1% MS-222 and embedded in 0.3% UltraPure agarose (Thermo Fisher Scientific, Waltham, MA) prepared in E3 medium containing

16 mg/ml MS-222. Measurements were performed at 28°C using a Leica DM IRBE inverted light microscope (Leica Microsystems) with a Hamamatsu C9300-221 high-speed CCD camera (Hamamatsu Photonics, Hamamatsu, Japan). Imaging was conducted at 150 frames per second using Hokawo 2.1 imaging software (Hamamatsu Photonics) for a period of 10 s (approximately 30 cardiac cycles). Heart rate measurements and contractility parameters were analyzed using ImageJ. Volumes were analyzed using ImageJ by drawing an ellipse on top of the ventricle at end-diastole and end-systole. Averages of three measurements per heart were determined. End diastolic and end systolic volume (EDV/ESV) were calculated by: $(4/3)*(\pi)*(\text{major axis}/2)*((\text{minor axis}/2)^2)$. Stroke volume (SV) by: EDV-ESV. Ejection fraction (EF) by: (SV/EDV)*100. Cardiac output (CO) by: SV*Heart rate.

## Whole-mount in situ hybridization

PTU-treated embryos were anesthetized in 0.1% MS-222 (Sigma-Aldrich, Ref: A5040) and fixed in 4% PFA for at least 12 hr at 4°C. *WISH* was performed as described in *Thisse et al., 1993*. Probes specific for *myl7, vhmc,* and *ahmc* were described in *Bonetti et al., 2014*. Probes specific for *c-myb, l-plastin, pu.1, gata1, ikaros, b-globin,* and *alas-2* were described in *Choorapoikayil et al., 2014*; *Hu et al., 2014*. Subsequently, embryos were mounted in 70% glycerol in PBS and imaged with Leica M165 FC stereomicroscope (Leica Microsystems). Images were processed in ImageJ (US National Institutes of Health, Bethesda, MD). Abundance of the probe signal was scored as low, mild, or high.

## Inhibitors treatment

PTU-treated embryos were incubated with either 0.15 µM of CI1040 (Sigma-Aldrich, Ref: PZ0181), 4 µM LY294002 (Sigma-Aldrich, Ref: L9908), or 10 µM of dexamethasone (Sigma-Aldrich, Ref: D4902) continuously from 48 hpf until 5 dpf in the same medium. At 5 dpf embryos were either fixed and half of the embryos was processed for WISH using probe specific for *c-myb* and the other half using probe specific for *l-plastin*, or snap-frozen and processed for RT-qPCR, while the live embryos in Tg(*mpx:GFP, mpeg:mCherry*) transgenic background were imaged by confocal microscopy.

## Confocal microscopy

All confocal imaging was performed on a Leica SP8 confocal microscope (Leica Microsystems). Embryos were mounted in 0.3% agarose. Live embryos were anesthetized in MS-222. Whole embryos were imaged using a 10× objective and z-stack step size of 3 µm, while the CHT area with 20× objective and z-stack step size of 1 µm. The number of CD41-GFP[low] cells was determined by imaging the CHT of the living 5 dpf old embryos of the Shp2[wt], Shp2[D61G/wt], and Shp2[D61G/D61G] siblings in the Tg(*cd41:GFP, kdrl:mCherry-CAAX*) transgenic background, while the number of CD41-GFP[high] cells was determined by imaging whole embryos, which were fixed for 2 hr in 4% PFA prior to imaging. To determine the number of mpx-GFP and mpeg-mCherry cells, whole live 5 dpf old embryos of the Shp2[wt], Shp2[D61G/wt], and Shp2[D61G/D61G] line in the Tg(*mpx:GFP, mpeg:mCherry*) transgenic background were imaged. Imaris V9.3.1 (Bitplane, Zurich, Switzerland) was used to reconstruct 3D images and count individual GFP and/or mCherry-positive cells.

## pHis3 staining

PTU-treated 5 dpf old Shp2[D61G] embryos in the Tg(*cd41:GFP, kdrl:mCherry-CAAX*) transgenic background were fixed in 2% PFA overnight and stained as described in *Choorapoikayil et al., 2012*. Primary pHis3 antibody (1:500 in blocking buffer, Abcam, Cambridge, UK, Ref: ab5176) and secondary GFP antibody (1:200 in blocking buffer, Aves Labs Inc, Tigard, OR, GFP-1010) were used. Embryos were mounted in 0.3% agarose, their CHT was imaged using the SP8 confocal microscope, and 3D images were subsequently reconstructed using Imaris.

## Acridine orange staining

PTU-treated embryos at 5 dpf were incubated in 5 µg/ml of Acridine orange (Sigma-Aldrich, Ref: A6014) in E3 medium, for 20 min at room temperature. They were then washed five times for 5 min in E3 medium, anesthetized in MS-222, and mounted in 0.3% agarose. Whole embryos were imaged with SP8 confocal microscope and 3D images were reconstructed using Imaris.

## Colony-forming assay

The CD41-GFP^low cell population isolated from CHTs of Shp2^wt, Shp2^D61G/wt embryos at 5 dpf was FACS-sorted; 250 µl of solution containing 2500 cells, media prepared as described in *Svoboda et al., 2016*, and 100 ng/ml of Gcsfa (gift from the Petr Bartunek lab, Institute of Molecular Genetics, Academy of Sciences of the Czech Republic v.v.i. Prague) was plated per well of a 96-well plate in a triplicate. Cells were grown in humidified incubators at 32°C, 5% $CO_2$. After 6 days, colonies were imaged and counted using the EVOS microscope (Thermo Fisher Scientific).

## Quantitative reverse transcription PCR

Zebrafish siblings of either Shp2^wt or Shp2^D61G/D61G genotypes were crossed to generate embryos of Shp2^wt and Shp2^D61G/D61G genotype. At 5 dpf, five embryos of the same condition were pooled and snap-frozen in liquid nitrogen. Total RNA was extracted from embryos using the TRIzol reagent according to the manufacturer's instructions (Thermo Fisher Scientific, Invitrogen, Waltham, MA, Ref: 15596018). cDNA synthesis was performed from 800 ng of RNA using the SuperScript III First-Strand kit (Thermo Fisher Scientific, Invitrogen, Waltham, MA, Ref: 12574018). RT-qPCR was performed with an CFX Connect Real-Time system (Bio-Rad Laboratories Inc, Hercules, CA) using FastStart Universal SYBR Green Master (ROX) (Roche, Basel, Switzerland) and 1:20 dilution of cDNA. Reaction mixtures were incubated for 10 min at 95°C, followed by 42 cycles of product amplification (15 s at 95°C and 1 min at 58°C). For each condition four samples were used in duplicate. Gene expression was normalized against the expression of *ef1a* for each mRNA. The following primers were used: *ef1a* forward, 5'-GAGAAGTTCGAGAAGGAAGC-3'; *ef1a* reverse, 5'-CGTAGTATTTGCTGGTCTCG-3'; *tnfa* forward, 5'-AGACCTTAGACTGGAGAGATGAC-3'; *tnfa* reverse, 5'-CAAAGACACCTGGCTGTAGAC-3'; *gcsfb* forward, 5'-AGAGAACCTACTGAACGACCT-3'; *gcsfb* reverse, 5'-CTTGAACTGGCTGAGTGGAG-3'; *il1b* forward, 5'-GAACAGAATGAAGCACATCAAACC-3'; *il1b* reverse, 5'-ACGGCACTGAATCCAC CAC-3'.

## Transplantation experiments

Zebrafish kidney marrow transplantation were performed as previously described (*Moore et al., 2016*; *Tang et al., 2014*). In short, tissues were isolated from donor Shp2^D61G/wt or wt animals in the Tg(*mpx-:GFP, mpeg:mCherry*) background following Tricaine (Western Chemical, Brussels, Belgium) overdose. Excised tissues from dissected fish are placed into 500 µl of 0.9× PBS + 5% FBS on a 10 cm Petri dish. Single-cell suspensions were obtained by maceration with a razor blade, followed by manual pipetting to disassociate cell clumps. Cells were filtered through a 40 µm Falcon cell strainer, centrifuged at 1000 *g* for 10 min, and resuspended to the $2 \times 10^7$ cells/ml. Five µl suspension containing $10^5$ kidney marrow cells were injected into the peritoneal cavity of each recipient fish using a 26s Hamilton 80366 syringe. Cellular engraftment was assessed at 0, 7, 14, 28 dpt by epifluorescence microscopy.

## Isolation of CD41-GFP^low cell population and single-cell RNA sequencing

Zebrafish siblings of either Shp2^wt or Shp2^D61G/D61G genotypes were crossed to generate embryos of Shp2^wt, Shp2^D61G/wt, and Shp2^D61G/D61G genotype. From 24 hpf onward, embryos were grown in PTU-containing medium. The CHTs of approximately 50 randomly selected Shp2^wt, Shpt^D61G/wt, and Shp2^D61G/D61G embryos in Tg(*cd41:GFP, kdrl:mCherry-CAAX*) transgenic background at 5 dpf were dissected and CHTs of the same genotype were pooled in Leibovitz medium (Thermo Fisher Scientific, Gibco, Waltham, MA, Ref: 11415049). After washing with PBS0, the CHTs were dissociated with TryplE (Thermo Fisher Scientific, Gibco, Waltham, MA, Ref: 12605036) for 45 min at 32°C. The resulting cell suspension was washed with PBS0, resuspended in PBS0 supplemented with 2 mM EDTA, 2% FCS, and 0.5 µg/ml DAPI (Sigma-Aldrich, Ref: D9542) and passed through a 40 µm Falcon cell strainer. DAPI staining was used to exclude dead cells. Cells with CD41-GFP^low-positive signal were subjected to FACS with an influx cytometer (BD Biosciences, San Jose, CA). Single-cell RNA sequencing was performed according to an adapted version of the SORT-seq (*Muraro et al., 2016*) with adapted primers described in *van den Brink et al., 2017*. In short, single cells were FACS-sorted, as described above, on 384-well plates containing 384 primers and mineral oil (Sigma-Aldrich). After sorting, plates were snap-frozen on dry ice and stored at –80°C. For amplification, cells were heat-lysed at 65°C followed by cDNA synthesis using the CEL-seq2 (*Hashimshony et al., 2016*) and robotic liquid handling platforms. After the second strand cDNA synthesis, the barcoded material was pooled into

libraries of 384 cells and amplified using in vitro transcription. Following amplification, the rest of the CEL-seq2 protocol was followed for preparation of the amplified cDNA library, using TruSeq small RNA primers (Illumina, San Diego, CA). The DNA library was paired-end sequenced on an Illumina Nextseq 500 (Illumina), high output, with a 1 × 75 bp Illumina kit (R1:26 cycles, index read: 6 cycles, R2:60 cycles).

## Data analysis of single-cell RNA sequencing

During sequencing, Read1 was assigned 26 base pairs and was used for identification of the Ilumina library barcode, cell barcode, and unique molecular identifier. Read2 was assigned 60 base pairs and used to map to the reference transcriptome of Zv9 *D. rerio*. Data was demultiplexed as described in *Grün et al., 2014*. Single-cell transcriptomics analysis was done using the RaceID3 algorithm (*Herman and Grün, 2018*), following an adapted version of the RaceID manual (https://cran.r-project.org/web/packages/RaceID/vignettes/RaceID.html) using R-3.5.2. In total 768 cells per genotype were sequenced for the datasets. After removing cells with less than 1000 UMIs and only keeping genes that were detected with at least 3 UMIs in 1 cell, 439 wt, 384 D61G/wt, and 543 D61G/D61G cells were left for further analysis. Batch effects observed for plates which were prepared on different days was removed using the scran function. Four major clusters and 5 minor clusters were identified. The minor clusters contained 130 cells in total and were excluded from further analysis for statistical reasons. Differential gene expression analysis was done as described in *Muraro et al., 2016*, with an adapted version of the DESseq2 (*Love et al., 2014*). GO term enrichment analysis for DEGs of each major cluster was performed using the DAVID Bioinformatics Resources 6.8 (https://david.ncifcrf.gov/). Trajectory inference and pseudotime analysis were done using Monocle3 (v1.0.0). The data was subset to include the major cell types: HSC-like cells, thrombocyte and erythrocyte progenitors, early myeloid progenitors, and monocyte/macrophage progenitors. The normalized counts, t-SNE coordinates, and clusters generated by RaceID3 were used for trajectory inference. To investigate the potential lineage differentiation path, the HSC-like cluster (center) was then selected as root for pseudotime analysis and used to order cells in pseudotime.

## RNA sequencing and differential gene expression analysis of the patient material

Libraries were prepared with TruSeq Stranded Total RNASample preparation kit (Illumina) according to supplier's recommendations. Briefly, the ribosomal RNA fraction was removed from 1 μg of total RNA using the Ribo-Zero Gold Kit (Illumina). Fragmentation was then achieved using divalent cations under elevated temperature to obtain approximately 300 bp pieces. Double strand cDNA synthesis was performed using reverse transcriptase and random primers, Illumina adapters were ligated, and cDNA library was PCR-amplified for sequencing. Paired-end 75b sequencing was then carried out on a HiSeq4000 (Illumina). Quality of reads was assessed for each sample using FastQC (http://www.bioinformatics.babraham.ac.uk/projects/fastqc/). A subset of 500,000 reads from each Fastq file was aligned to the reference human genome hg19/GRCh37 with tophat2 to determine insert sizes with Picard. Full Fastq files were aligned to the reference human genome hg19/GRCh37 with tophat2 (-p 24r 150g 2 `--library-type` fr-firststrand) . We removed reads mapping to multiple locations. STAR was used to obtain the number of reads associated with each gene in the Gencode v26lift37 annotation (restricted to protein-coding genes, antisense, and lincRNAs). Raw counts for each sample were imported into R statistical software using the Bioconductor DESeq2 package. Extracted count matrix was normalized for library size and coding length of genes to compute TPM expression levels. Differential gene expression analysis was performed using the Bioconductor limma package and the voom transformation. To improve the statistical power of the analysis, only genes expressed in at least one sample (TPM > 0.3) were considered. A qval threshold of <0.05 and a minimum fold change of 1.2 were used to define DEGs. Due to the imbalance between male and female samples, sex differences were adjusted with the function model.matrix from the stats package. RNA sequencing data analysis was performed with Galileo v1.4.4, an interactive R shiny application. Representations of the gene expression levels were performed with the library ggplot2 and R 4.0.2.

## Pathway enrichment analysis (GSEA) and PCA

GSEA was performed by clusterProfiler::GSEA function using the fgsea algorithm. Gene list from the differential analysis was ordered by decreasing log2 (fold change) values. Hallmark classes gene sets from the MSigDB v7.2 database were selected keeping only gene sets defined by 10–500 genes. The p-values were adjusted by the Benjamini-Hochberg (FDR) procedure. The custom zebrafish signature was built from the human orthologous genes of the 100 most overexpressed zebrafish genes based on the log2 (fold change) values derived from the monocyte/macrophage progenitor cluster of the zebrafish HSPCs single-cell sequencing dataset. The Bioconductor edgeR package was used to import raw counts into R, and compute normalized log2 CPM (counts per millions of mapped reads) using the TMM (weighted trimmed mean of M-values) as normalization procedure. The normalized expression matrix from the 100 most overexpressed genes (based on log2FC in the zebrafish dataset) was used to classify the samples according to their gene expression patterns using PCA. PCA was performed by FactoMineR::PCA function with 'ncp = 10, scale.unit = FALSE' parameters.

## Statistical analysis

Data was plotted in GraphPad Prism 7.05 (GraphPad Software Inc, San Diego, CA), except for the violin plots of gene expression, which were plotted in R using ggplot2 (*Wickham, 2009*). Statistical difference analysis was performed using the one-way ANOVA supplemented by Tukey's HSD test in GraphPad Prism 7.05, except for the gene expression differences in single-cell RNA sequencing data, where t-test was performed in Rstudio 1.1.463 (Rstudio, Boston, MA), and the treatment WISH experiments, where chi-squared test was performed in GraphPad Prism 7.05. Significant difference was considered when $p < 0.05$ (*$p < 0.05$, **$p < 0.01$, ***$p < 0.001$, NS = non-significant).

## Acknowledgements

We thank the laboratory of Niels Geijsen (Hubrecht Institute) for providing us with Cas9 protein, Petr Bartunek and Olga Machanova (Institute of Molecular Genetics, Academy of Sciences of the Czech Republic v.v.i. Prague), and Geert Wiegertjes (University Wageningen) for cytokines and carp serum, Stefan van der Elst for assistance with FACS, Hubrecht Institute animal caretakers for animal support and Single Cell Discoveries, and Chloé Baron for support with single-cell RNA sequencing. This work was supported by E-Rare grant NSEuroNet (JdH and HC), EJPRD grant NSEuroNet (JdH and HC), KWF Dutch Cancer Society grant (JdH), NIH grants R01CA211734 (DML) and R24OD016761 (DML), the MGH Research Scholar Award (DML), and Alex Lemonade Stand Foundation (CY).

## Additional information

### Funding

| Funder | Grant reference number | Author |
|---|---|---|
| European Commission | ERARE NSEURONET | Helene Cave Jeroen den Hertog |
| European Commission | EJPRD NSEURONET | Helene Cave Jeroen den Hertog |
| KWF Kankerbestrijding | 12829 | Jeroen den Hertog |
| NIH Office of the Director | R01CA211734 | David M Langenau |
| NIH Office of the Director | R24OD016761 | David M Langenau |
| MGH Research Scholar Award | | David M Langenau |
| Alex Lemonade Stand Foundation | | Chuan Yan |

The funders had no role in study design, data collection and interpretation, or the decision to submit the work for publication.

## Author contributions
Maja Solman, Conceptualization, Data curation, Formal analysis, Investigation, Visualization, Writing - original draft, Writing – review and editing; Sasja Blokzijl-Franke, Chuan Yan, Qiqi Yang, Data curation, Formal analysis, Investigation, Methodology, Writing – review and editing; Florian Piques, Formal analysis, Investigation, Methodology, Writing – review and editing; Marion Strullu, Formal analysis, Investigation, Methodology; Sarah M Kamel, Investigation, Methodology, Resources, Writing – review and editing; Pakize Ak, Formal analysis, Investigation; Jeroen Bakkers, David M Langenau, Hélène Cavé, Conceptualization, Supervision, Writing – review and editing; Jeroen den Hertog, Conceptualization, Funding acquisition, Project administration, Supervision, Writing - original draft, Writing – review and editing

## Author ORCIDs
Sarah M Kamel (ID) http://orcid.org/0000-0003-4424-9732
Jeroen Bakkers (ID) http://orcid.org/0000-0002-9418-0422
David M Langenau (ID) http://orcid.org/0000-0001-6664-8318
Hélène Cavé (ID) http://orcid.org/0000-0003-2840-1511
Jeroen den Hertog (ID) http://orcid.org/0000-0002-8642-8088

## Ethics
All children's samples were obtained after parents had given their written informed consent. Experiments were approved by the institutional review board of the French Institute of Health and Medical Research (INSERM) (IORG0003254) in accordance with the Helsinki declaration. Healthy children bone marrows were obtained from intrafamilial BM transplantation donors and used with the approval of the Institutional Review Board of "Hôpitaux Universitaires Paris Nord Val-de-Seine," Paris 7 University, AP-HP, (IRB: 00006477), in accordance with the Helsinki declaration.

All procedures involving experimental animals were approved by the animal experiments committee of the Royal Netherlands Academy of Arts and Sciences (KNAW), Dierexperimenten commissie protocol HI18-0702, and performed under the local guidelines in compliance with national and European law.

## Decision letter and Author response
Decision letter https://doi.org/10.7554/eLife.73040.sa1
Author response https://doi.org/10.7554/eLife.73040.sa2

---

# Additional files

## Supplementary files
• Transparent reporting form

• Supplementary file 1. List of differentially expressed genes (DE) and enriched gene ontology terms (GO) in each of the 4 major clusters of zebrafish HSPCs.

• Supplementary file 2. Characteristics of JMML PTPN11 mutated patients and healty subjects.

• Supplementary file 3. List of differentially expressed genes (DE) in JMML patient vs. normal bone marrow derived HSPCs.

• Supplementary file 4. GSEA analysis of differentially expressed genes in JMML patient derived HSPCs.

## Data availability
Sequencing data has been deposited to GEO under accession codes GSE167787 and GSE183252 Figure 1-Source Data 1, Figure 3-Source Data 1 and Figure 5-Source Data 1-3 contain the source data for the respective figures.

*Continued on next page*

The following datasets were generated:

| Author(s) | Year | Dataset title | Dataset URL | Database and Identifier |
|---|---|---|---|---|
| Solman M, Blokzijl-Franke S, Yan C, Yang Q, Kamel SM, Bakkers J, Langenau DM, den Hertog J | 2021 | Inflammation potentiates JMML-like blood defects in Shp2 mutant Noonan syndrome | http://www.ncbi.nlm.nih.gov/geo/query/acc.cgi?acc=GSE167787 | NCBI Gene Expression Omnibus, GSE167787 |
| Cavé H, Piques F, Strullu M, Arfeuille C, Caye A | 2021 | Inflammatory response in HSPC of PTPN11-mutated JMML | http://www.ncbi.nlm.nih.gov/geo/query/acc.cgi?acc=GSE183252 | NCBI Gene Expression Omnibus, GSE183252 |

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
