## [Editor Report]

The authors of this paper model the D61G mutation in the gene PTPN11 that encodes the protein-tyrosine phosphatase SHP2 in zebrafish, creating a model consistent with the human Noonan syndrome (NS), which is predisposed to juvenile myelomonocytic leukemia (JMML) and myeloproliferative neoplasm (MPN)-like syndrome. The study nicely provides a new model that can be used as the basis for future studies in the field. Because the mutant variably displays phenotypes along a spectrum from NS to MPN, different researchers can choose to focus on this as they see fit.

---

## [Decision Letter]

**Decision letter after peer review:**

Thank you for submitting your article "Inflammatory response in hematopoietic stem and progenitor cells triggered by activating SHP2 mutations potentiates leukemogenesis" for consideration by *eLife*. Your article has been reviewed by 3 peer reviewers and the evaluation has been overseen by Richard White as the Senior Editor and Reviewing Editor. The reviewers have opted to remain anonymous.

Essential revisions:

The detailed revisions are explained below, but I wanted to highlight two important major things that were common across reviewers:

1) Concordance between the fish and human diseases, especially in regards to HSPC vs macrophage/monocyte populations. This issue was brought up by several reviewers, as detailed below. As it stands, it makes the title of the paper confusing.

2) A better way to address the effect of the anti-inflammatory drugs you have chosen in terms of staining for marker genes by in situ or via transgenic reporters of the various cells affected. This was also raised by several reviewers and needs clarification.

*Reviewer #1 (Recommendations to the Authors):*

1. Is there evidence of splenomegaly, as is observed consistently in children with JMML, in the fish?

2. Onlly two genes (pu.1 and alas-2) are fully evaluated for myeloid bias.

3. Figure 3c does not show granulocytic lineage. Please explain.

4. There is not much difference between heterozygous and homozygous D61G fish in terms of phenotype penetrance or gene expression. Please explain.

5. What do the heterozygous or homozygous D61G fish die from?

6. The title overstates the data presented, including the statement of "potentiating leukemogenesis".

*Reviewer #2 (Recommendations for the authors):*

The manuscript is well written and easy to read. I have some comments of parts that were incomplete or not very clear to me:

1. It is not clear if this is a dominant or recessive mutation in patients.

2. There is no mention how SHP2 is conserved between species (human vs zebrafish). A figure showing the amino acid conservation, especially the D61 amino acid would be helpful.

3. In zebrafish there are two orthologues of SHP2: Shp2a and Shp2b. They authors do not address what is the difference between them and why they chose Sh2a. Are they expressed in different times during development? Or in different tissues? Do they see the same phenotype in Shp2b-/-?

4. Figure 1J. The mutants that showed a severe phenotype after 24 wpf, do they have any hematological defect? Are they fertile?

5. Figure 1K. There is not a legend. Is it statistical different?

6. Figure 2. All images are wt and D61G/wt. Why not showing the D61G/G61G images?

7. Figure 3A. Is each larvae genotyped before flow experiments? Do the larvae selected presented a phenotype at 5 dpf as described in lines 112-113. Is it a pool of 50 larvae per genotype?

8. In material and methods is not included the anti-inflammatory treatments. It would be useful to include drug's company, concentrations, if change of media daily… etc. It would be interesting to see the effect of these drugs in neutrophils and macrophages using the transgenic lines Tg(mpx:GFP) and Tg(mpeg:mCherry).

*Reviewer #3 (Recommendations for the authors):*

Overall, this is an interesting, well written and clear manuscript. The mutant is a very nice tool that can be use by others in the field, and makes good use of complementary fish and human samples. I find some of the RNA-seq comparisons confusing, which dovetails with a lack of clear mechanism. Although the paper is clearly set up as a "model" paper, rather than a mechanism paper, some of this confusion hampers the ability to make conclusions and would be strengthened by a few things:

1) Fish vs human

Perhaps the most confusing aspect of the study is where and when the inflammatory signature occurs in fish vs humans. In the fish scRNA, they mainly seem to identify this in the monocyte/macrophage clusters, yet the paper title and abstract implies that this occurs in the HSPC cluster. This is very aptly demonstrated in the human samples, where that signature seems robust. However, it is not clear from the data whether this signature truly exists in the fish HSPCs or is it restricted to the monocyte/macrophages? The authors need to show the pathway analysis for both clusters in detail, and then discuss why the fish might be different than the human (perhaps it is not).

2) The cell of origin

The above leads to some confusion of what you think the cell of origin is in fish vs. human JMML/MPN phenotypes. Do you they diseases, although similar, arise in different populations? Do you have data to suggest this? One way of getting at this would be to do something like RNA Velocity in the scRNA data, since this may tell you where the inflammatory signature comes from. Transplantation of the HSPC vs monocyte/macrophage populations from the fish would be even better if that were technically feasible (this is not a required experiment). Depending on this data, a much clearer description of the cell of origin needs to be made.

3) The mechanism

I do not expect a detailed mechanism in this type of paper. Yet the RNA-seq is a real opportunity to at least suggest potential mechanisms. I am not convinced this is a cell autonomous effect, and I would think that there could be cell-cell interactions that help explain this. One idea would be to run something like NicheNet or CellPhoneDB on the HSPC vs. monocyte/macrophage populations from the scRNA-seq, which would at least propose potential cell-cell interaction mechanisms that could be explore by other researchers down the road.

---

## [Author Response]

Essential Revisions:The detailed revisions are explained below, but I wanted to highlight two important major things that were common across reviewers:1) Concordance between the fish and human diseases, especially in regards to HSPC vs macrophage/monocyte populations. This issue was brought up by several reviewers, as detailed below. As it stands, it makes the title of the paper confusing.

A direct comparison between fish and human cannot be made, because the type of data is different (single-cell RNAseq for fish and bulk RNAseq for human) and the source and stage of the material is different. Embryonic CD41-GFP^low^ positive HSPCs were isolated from zebrafish embryos and postnatal FACS-sorted HSPCs from human patients. The human material was sorted based on the immunophenotypic signatures and included HSCs and progenitors. We have included a remark about the identity of the human cells that were used in the Results section (p. 16, line 305). In zebrafish, such an approach to isolate distinct compartments of HSPCs is not available, because of the lack of appropriate antibodies. Nevertheless, CD41-GFP^low^ cells have been reported to represent the HSPCs population in zebrafish and these cells were analyzed by single-cell RNA sequencing, which confirmed their identity as HSC-like cells, early myeloid progenitors, monocyte/macrophage progenitors and thrombocyte/ erythroid progenitors. Hence, the fish and human material both included HSCs and progenitors and we have indicated this in the text of the revised manuscript. The finding that fish and human HSPCs both show an inflammatory response is striking. We advanced the single cell RNA sequencing analysis and established that the proinflammatory response is evoked during early differentiation of monocyte/macrophage progenitors in Shp2D61G zebrafish HSPCs (see new Figure 4C-E and Figure 4—figure supplement 1 of the revised manuscript).

In response to the specific comment of reviewer 1 regarding leukemogenesis in the title, we have changed the title into: “Inflammatory response in hematopoietic stem and progenitor cells triggered by activating SHP2 mutations evokes blood defects”.

2) A better way to address the effect of the anti-inflammatory drugs you have chosen in terms of staining for marker genes by in situ or via transgenic reporters of the various cells affected. This was also raised by several reviewers and needs clarification.

We have extended our analysis of the effects of the anti-inflammatory drug, dexamethasone, from the in situ hybridization experiments in the original manuscript to analysis of the number of neutrophils and macrophages in Tg(*mpx:GFP, mpeg:mCherry*) transgenic zebrafish embryos and RT-qPCR experiments for inflammatory genes (see new Figure 6D-G of the revised manuscript). All parameters indicated that dexamethasone rescued the effects of the Shp2-D61G mutation, by directly affecting the observed myeloid lineage expansion and decreasing proinflammatory gene expression. We conclude that the inflammatory response may have a causal role in the observed blood defects.

Reviewer #1 (Recommendations to the Authors):1. Is there evidence of splenomegaly, as is observed consistently in children with JMML, in the fish?

Zebrafish spleen has not been studied during embryonic development, and thus, due to lack of markers we have not analyzed the presence of splenomegaly during zebrafish embryogenesis.

2. Onlly two genes (pu.1 and alas-2) are fully evaluated for myeloid bias.

Besides the pu.1 and alas2 markers, which were used to show the myeloid bias already during the myeloid and erythroid progenitors differentiation, distinct blood lineages were investigated either by WISH or in transgenic lines and are presented throughout the manuscript. All of these markers support the myeloid bias in Shp2-D61G zebrafish embryos. In particular:

– Figure 2. E,F white blood cells investigated by WISH using *l-plastin* marker

– Figure 2 G-I macrophages and neutrophils investigated in the tg(*mpeg:mCherry*, *mpx:GFP*) transgenic background

– Figure 2—figure supplement 1D,E *ikaros* as a marker for lymphocytes

– Figure 2—figure supplement 1F thrombocytes investigated in the Tg(*cd41:GFP*) transgenic background, in which CD41-GFP^high^ represents thrombocytes

– Figure 2—figure supplement 1G,H *globin* as a marker for erythrocytes

Next to these markers, the myeloid bias in mutant embryos is based on the single cell RNAseq experiments and panels of genes that are representative for the different lineages (Figure 3—figure supplement 1).

3. Figure 3c does not show granulocytic lineage. Please explain.

The granulocytic lineage is part of the early myeloid progenitor cluster. Our analysis did not separate the cells of granulocyte progenitors into a separate cluster. In Figure 3-supplement figure 1B, we plotted neutrophil-specific markers, showing a subcluster of more differentiated neutrophil progenitors.

4. There is not much difference between heterozygous and homozygous D61G fish in terms of phenotype penetrance or gene expression. Please explain.

Indeed, phenotypes are similar between heterozygous and homozygous fish. Phenotypes appear to be more pronounced in homozygous embryos than in heterozygous embryos, which may reflect that Shp2-D61G is a gain-of-function mutation that – like in human patients – acts in a dominant manner.

5. What do the heterozygous or homozygous D61G fish die from?

We have not investigated this in detail, because Shp2-D61G fish do not die prematurely.

6. The title overstates the data presented, including the statement of "potentiating leukemogenesis".

We have changed the title to “Inflammatory response in hematopoietic stem and progenitor cells triggered by activating SHP2 mutations evokes blood defects”.

Reviewer #2 (Recommendations for the authors):The manuscript is well written and easy to read. I have some comments of parts that were incomplete or not very clear to me:1. It is not clear if this is a dominant or recessive mutation in patients.

Noonan syndrome is a dominantly inherited developmental disorder. We included “heterozygous” in the first sentence of the introduction to underline this (p. 3, line 49):

“A broad spectrum of heterozygous germline activating mutations in the tyrosine phosphatase SHP2, encoded by PTPN11 has been found to cause Noonan syndrome (NS), a dominantly inherited developmental disorder from the RASopathy group affecting 1:1,500 individuals. NS is characterized by a systemic impact on development, most commonly resulting in short stature, congenital heart defects and specific craniofacial characteristics (Rauen, 2013; Tajan et al., 2018).”

Moreover, we included a statement in the last section of the introduction to make it clear that the D61G mutation is a dominantly inherited mutation (p. 4, line 88):

“To better assess the link between dysregulated SHP2 and myeloproliferation in the context of NS, we developed and characterized a novel genetic zebrafish model of NS with Shp2-D61G mutation, a dominantly inherited NS-associated mutation that is most frequently associated with NS/JMML-like MPN in human patients (Strullu et al., 2014; Tartaglia et al., 2001).”

2. There is no mention how SHP2 is conserved between species (human vs zebrafish). A figure showing the amino acid conservation, especially the D61 amino acid would be helpful.

The overall sequence homology between human SHP2 and zebrafish Shp2a is 91%. It is noteworthy that the D61 amino acid is located in an absolutely conserved region of the protein. The N-terminal 119 amino acids of SHP2 and Shp2a are identical. We have included a statement about sequence conservation in the text (p. 5, line 114-116) and we have included a sequence alignment in the Supplementary Material (Figure 1—figure supplement 1).

3. In zebrafish there are two orthologues of SHP2: Shp2a and Shp2b. They authors do not address what is the difference between them and why they chose Sh2a. Are they expressed in different times during development? Or in different tissues? Do they see the same phenotype in Shp2b-/-?

We have studied the two zebrafish orthologues of SHP2, Shp2a and Shp2b, in detail before (Bonetti et al., 2014, PLoS ONE 9, e94884). Loss of functional Shp2a is embryonic lethal, whereas Shp2b is dispensable and Shp2b-/- fish are viable and fertile. This is why we targeted Shp2a for the D61G gain-of-function mutation. We have included a statement to this effect in the text (p. 5, line 108-111). It is noteworthy that Shp2b is also dispensable in heterozygous and homozygous Shp2a-D61G background, in that the Shp2a-D61G phenotypes were indistinguishable between Shp2b+/+ and Shp2b-/- backgrounds. We have not included this in the manuscript, because it is beyond the scope of the paper.

4. Figure 1J. The mutants that showed a severe phenotype after 24 wpf, do they have any hematological defect? Are they fertile?

We are currently investigating this subset of fish in detail. Because the phenotype is not fully penetrant and because it is not evident that this phenotype relates to NS in human patients, we did not include a detailed analysis in this manuscript. We included a statement to this effect in the revised manuscript (p. 6, line 156-157).

5. Figure 1K. There is not a legend. Is it statistical different?

We have corrected the omission and included the legend in Figure 1K. We performed a statistical comparison between heterozygous, homozygous and wild type embryos using a two-way ANOVA complemented with Tukey’s HSD test and found that the differences were significant (p < 0.001).

6. Figure 2. All images are wt and D61G/wt. Why not showing the D61G/G61G images?

We only show images of heterozygous embryos because these are representative of human patients, who are heterozygous too. We do show the quantifications of homozygous embryos and it is evident that homozygous embryos show slightly stronger phenotypes than heterozygous embryos. We have included images of the homozygous D61G embryos as a separate figure for the reviewer to evaluate.

7. Figure 3A. Is each larvae genotyped before flow experiments? Do the larvae selected presented a phenotype at 5 dpf as described in lines 112-113. Is it a pool of 50 larvae per genotype?

For the flow experiments in Figure 3 and for the newly added RT-qPCR experiments in Figure 6G, crosses were set up to generate clutches of embryos with uniform genotypes, Shp2*^wt^*, Shp2*^D61G/wt^* and Shp2*^D61G/D61G^*, by crossing wild type, wild type and homozygous and incrossing homozygous fish, respectively. Whereas genotyping by finclipping as early as 3 dpf is feasible (Figure 1K), it might have severe adverse effects on the expression data. To avoid large differences in genetic backgrounds, the adult fish that were used for these crosses were siblings from the same family. We used a pool of 50 embryos for each sample in the flow experiments. We have included a statement about how these pools of embryos were generated in the Materials and methods section (p. 27, line 545-547 and p. 30, line 633-634).

8. In material and methods is not included the anti-inflammatory treatments. It would be useful to include drug's company, concentrations, if change of media daily… etc. It would be interesting to see the effect of these drugs in neutrophils and macrophages using the transgenic lines Tg(mpx:GFP) and Tg(mpeg:mCherry).

We have reviewed the text and made appropriate changes to clarify the drug treatments (p. 29, line 586-594). We have determined the effect of dexamethasone on the number of neutrophils and macrophages (new Figure 6D-F). Dexamethasone largely rescued the increase in number of neutrophils and macrophages that was observed in Shp2-D61G mutant embryos.

Reviewer #3 (Recommendations for the authors):Overall, this is an interesting, well written and clear manuscript. The mutant is a very nice tool that can be use by others in the field, and makes good use of complementary fish and human samples. I find some of the RNA-seq comparisons confusing, which dovetails with a lack of clear mechanism. Although the paper is clearly set up as a "model" paper, rather than a mechanism paper, some of this confusion hampers the ability to make conclusions and would be strengthened by a few things:1) Fish vs humanPerhaps the most confusing aspect of the study is where and when the inflammatory signature occurs in fish vs humans. In the fish scRNA, they mainly seem to identify this in the monocyte/macrophage clusters, yet the paper title and abstract implies that this occurs in the HSPC cluster. This is very aptly demonstrated in the human samples, where that signature seems robust. However, it is not clear from the data whether this signature truly exists in the fish HSPCs or is it restricted to the monocyte/macrophages? The authors need to show the pathway analysis for both clusters in detail, and then discuss why the fish might be different than the human (perhaps it is not).

A direct comparison between fish and human cannot be made, because the type of data is different (single-cell RNAseq for fish and bulk RNAseq for human) and the source and stage of the material is different. Embryonic CD41-GFP^low^ positive HSPCs were isolated from zebrafish embryos and postnatal FACS-sorted HSPCs from human patients. The human material was sorted based on the immunophenotypic signatures and included HSCs and progenitors. We have included a remark about the identity of the human cells that were used in the Results section (p. 17, line 306). In zebrafish, such an approach to isolate distinct compartments of HSPCs is not available, because of the lack of appropriate antibodies. Nevertheless, CD41-GFP^low^ cells have been reported to represent the HSPCs population in zebrafish and these cells were analyzed by single-cell RNA sequencing, which confirmed their identity as HSC-like cells, early myeloid progenitors, monocyte/macrophage progenitors and thrombocyte and erythroid progenitors. Hence, the fish and human material both included HSCs and progenitors. The finding that fish and human HSPCs both show an inflammatory response is striking.

2) The cell of originThe above leads to some confusion of what you think the cell of origin is in fish vs. human JMML/MPN phenotypes. Do you they diseases, although similar, arise in different populations? Do you have data to suggest this? One way of getting at this would be to do something like RNA Velocity in the scRNA data, since this may tell you where the inflammatory signature comes from. Transplantation of the HSPC vs monocyte/macrophage populations from the fish would be even better if that were technically feasible (this is not a required experiment). Depending on this data, a much clearer description of the cell of origin needs to be made.

As described in our public response to reviewer 3: We have used the single cell sequencing dataset of zebrafish HSPCs to determine when pro-inflammatory gene expression was evoked. We used Monocle trajectory inference analysis and found that differentiation started in HSCs-like cells and progressed in two directions in pseudotime, towards monocyte/macrophage progenitors in one direction and towards thrombocyte and erythroid progenitors in the other direction. The analysis showed that proinflammatory genes are evoked during early differentiation of monocyte/macrophage progenitor cells specifically in the Shp2*^D61G^* mutants. We have included this analysis in the new panels C-E in Figure 4 and Figure 4 —figure supplement 1 of the revised version of the manuscript and accompanying textual changes in the Results section. Treatment with an anti-inflammatory agent largely rescued the inflammatory response and the observed blood defects in Shp2*^D61G^* mutants, suggesting a causal role for the inflammatory response. Thus, according to these analyses, the cells of origin are HSPCs with monocyte/ macrophage progenitor characteristics.

3) The mechanismI do not expect a detailed mechanism in this type of paper. Yet the RNA-seq is a real opportunity to at least suggest potential mechanisms. I am not convinced this is a cell autonomous effect, and I would think that there could be cell-cell interactions that help explain this. One idea would be to run something like NicheNet or CellPhoneDB on the HSPC vs. monocyte/macrophage populations from the scRNA-seq, which would at least propose potential cell-cell interaction mechanisms that could be explore by other researchers down the road.

This is an interesting suggestion, which however would require another RNAseq experiment. We are not able to extract this type of information from the single cell RNAseq dataset that we have generated. Whereas we do not provide profound insights into the underlying mechanism, we do provide evidence that the inflammatory response may have a causal effect on the observed blood defects.